# **LC-PLM**: Long-context Protein Language Model

## Abstract

Self-supervised training of language models (LMs) has seen great success for protein sequences in learning meaningful representations and for generative drug design. Most protein LMs are based on the Transformer architecture trained on individual proteins with short context lengths. Such protein LMs cannot extrapolate to longer proteins and protein complexes well. They also fail to account for the underlying biological mechanisms carried out by biomolecular interactions and dynamics i.e., proteins often interact with other proteins, molecules, and pathways in complex biological systems. In this work, we propose **LC-PLM** based on an alternative protein LM architecture, **BiMamba-S**, built off selective structured state-space models, to learn high-quality universal protein representations at the amino acid token level using masked language modeling. We also introduce its graph-contextual variant, **LC-PLM-G**, which contextualizes protein-protein interaction (PPI) graphs for a second stage of training. **LC-PLM** demonstrates favorable neural scaling laws, better length extrapolation capability, and a 7% to 34% improvement on protein downstream tasks than Transformer-based ESM-2. **LC-PLM-G** further trained within the context of PPI graphs shows promising results on protein structure and function prediction tasks. Our study demonstrates the benefit of increasing the context size with computationally efficient LM architecture (e.g. structured state space models) in learning universal protein representations and incorporating molecular interaction context contained in biological graphs.

## 1 Introduction

Most biological sequences are derived from genomes, which are long DNA sequences: human chromosomes range from 50 to 300 million base pairs. The protein-coding regions, which can be considered as the translated substrings of the genome, are relatively shorter (the majority are < 3,000 amino acids), albeit with a few exceptions, such as Titin, composed of 34K amino acids. The prevalent protein language models (pLMs), e.g. ESM-2 (Lin et al., 2023), choose 1024 as the context length as it fits 97.4% of proteins. However, it does not natively support tasks that require long-range context windows to reason over multiple related sequences, such as genomic interactions, protein-protein interactions (PPI), protein function prediction, and 3D structure prediction of long proteins and protein complexes. Another challenge for modeling long-range biological contexts lies in their non-sequential nature. For instance, the useful context for genomic interactions and PPIs often span across regions from different chromosomes, and capturing information within an LM of such interactions usually requires biomedical knowledge graphs for good performance on these tasks (Kovács et al., 2019; Sousa et al., 2024).

Large LMs including those trained on protein sequences, are predominantly based on the Transformer (Vaswani et al., 2017) with multi-head attention. Despite its state-of-the-art performance on virtually all types of data modalities (texts, vision, audio, etc.), it suffers from quadratic time and space complexity due to the lengths of the input sequences. Additionally, transformer models are known to have poor length extrapolation quality and do not achieve the same level of performance when evaluated on sequences longer than seen during pretraining. Recent work in alternative architectures such as convolutional (e.g. Hyena (Poli et al., 2023)) and selective structured state space models (SSMs) (e.g. Mamba (Gu & Dao, 2024)) have demonstrated competitive performance and preferable scaling properties on long context compared to Transformers and extensions including linear attention approximation variants (Katharopoulos et al., 2020; Zhai et al., 2021; Peng et al., 2023). Although recent studies have leveraged these novel architectures to train LMs for DNA sequences (Nguyen et al., 2024b;a; Schiff et al., 2024), studies examining their feasibility as protein LMs are limited.

There is also a research gap on how to effectively leverage the long-context capability of these architectures to model graphs of sequences i.e., how to leverage PPI graphs to improve LM's ability to reason across interacting (related) proteins.

In this work, we explore alternative architectures based on Mamba to improve the long-context capability of pLMs. We train a long-context pLM (**LC-PLM**) using bidirectional Mamba with shared projection layers (**BiMamba-S**) on protein sequences from `UniRef50` with masked language modeling (MLM) objective. Results show favorable neural scaling laws, length extrapolation properties on `UniRef90`, and better

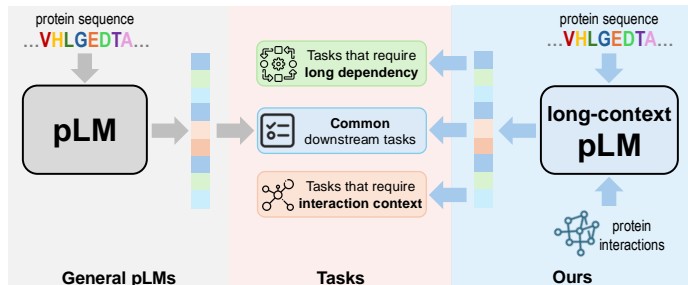

Figure 1: Our model enables long-context capability, length extrapolation ability, better neural scaling law, and interaction context-aware inference.

downstream task performance on `TAPE` (Rao et al., 2019) and `ProteinGym` (Notin et al., 2024) than its Transformer counterpart, namely ESM-2. Such long-context and length extrapolation properties facilitate and improve structure prediction of long proteins and protein complexes from `CASP14`, `CASP15-multimers`, and `Benchmark2`. Next, we train a graph-contextualized variant **LC-PLM-G**, which uses a proposed novel second-stage training strategy to leverage the long-context capabilities of **BiMamba-S** to encode useful information from interaction graphs (e.g., PPI). Trained on sampled random walks that are composed of sequences of proteins, **LC-PLM-G** improves performance on remote homology prediction, node-level protein function prediction (`ogbn-proteins`), and link-level PPI prediction (`ogbl-ppa`) (Hu et al., 2020). Our contributions can be summarized into three folds as follows:

- We develop a long-context pLM (**LC-PLM**) with an alternative architecture based on a more sample & compute-efficient bidirectional Mamba architecture with shared projection layers (**BiMamba-S**) pretrained on `UniRef50` with MLM objective.

- We demonstrate that **LC-PLM** has improved length extrapolation capabilities, favorable scaling laws, and achieved a 7% to 34% improvement on downstream tasks (e.g. protein structure prediction (`CASP15-multimers`, `CASP14`, `Benchmark2`), tasks in `TAPE` and `ProteinGym`) compared to ESM-2, especially for longer proteins and protein complexes.

- To encode biological interaction information, we propose a novel second-stage training based on random walks over graphs to extend the long-context capabilities of **LC-PLM** to leverage the PPI graph context. We demonstrate its effectiveness in capturing graph-contextual information on remote homology detection (`TAPE`), protein function prediction (`ogbn-proteins`), and PPI link prediction (`ogbl-ppa`).

## 2 RELATED WORKS

### 2.1 LONG-CONTEXT LMs AND STATE SPACE MODELS

Since their introduction, Transformers (Vaswani et al., 2017) with multi-head attention have been successfully applied in many different applications in natural language and computer vision. However, while being relatively straightforward to scale the number of parameters, Transformer models have a quadratic dependence on the context length during training and are linear at inference time, making them expensive to scale to long context. Alternative to Transformers, Recurrent Neural Networks (RNNs) (Hochreiter, 1991; Bengio et al., 1994; Hochreiter & Schmidhuber, 1997) scale more favorably with context length and have linear dependency at training time and constant at inference time. However, generic non-linear RNNs cannot be parallelized on modern hardware due to the sequential nature of their gradient update rule (Bengio et al., 1994).

To improve RNNs scalability on modern hardware, recent works on SSMs (Gu et al., 2021; Fu et al., 2022; Gu & Dao, 2024) propose to linearize RNNs dynamics and use efficient hardware-aware algorithms. A notable example is Mamba (Gu & Dao, 2024), which leverages the associative scan to

efficiently process arbitrarily long sequences in linear time, and Mamba-2 (Dao & Gu, 2024) that greatly improves over Mamba by implementing SSM layers using structured matrix multiplications to better leverage modern Tensor cores.

To further harvest the benefits of SSM and Transformer primitives, hybrid models have been proposed in Zancato et al. (2024); Lieber et al. (2024); Arora et al. (2024); De et al. (2024); Botev et al. (2024); Waleffe et al. (2024). There are also efforts trying to extend Mamba models to graph data (Wang et al., 2024; Behrouz & Hashemi, 2024). However, unlike our `LC-PLM-G`, which learns token-level protein representations within graph context from the graph of sequences, they focus on learning node/graph-level representations that only work for generic graph tasks where nodes do not contain sequences (see Table 1).

| Method | Universality | Fine granularity | Long-context capability *Handleability* & *Performance* | | Graph context | Large-scale model |
|---|---|---|---|---|---|---|
| ProtGPT (2022) | ✓ | ✓ | ✗ | ✗ | ✗ | ✗ |
| ESM-2 (2023) | ✓ | ✓ | ✗ | ✗ | ✗ | ✓ |
| CARP (2024) | ✓ | ✓ | ✓ | ✗ | ✗ | ✓ |
| ProtHyena (2024) | ✓ | ✓ | ✓ | ✗ | ✗ | ✗ |
| PoET (2024) | ✗ | ✓ | ✗ | ✗ | ✗ | ✗ |
| ProtMamba (2024) | ✗ | ✓ | ✓ | ✗ | ✗ | ✗ |
| PTM-Mamba (2024) | ✗ | ✓ | ✓ | – | ✗ | ✓ |
| Graph-Mamba (2024) | ✗ | ✗ | ✓ | – | ✓ | ✗ |
| GMN (2024) | ✗ | ✗ | ✓ | – | ✓ | ✗ |
| **LC-PLM** | ✓ | ✓ | ✓ | ✓ | ✗ | ✓ |
| **LC-PLM-G** | ✓ | ✓ | ✓ | ✓ | ✓ | ✓ |

Table 1: Comparison of `LC-PLM` and `LC-PLM-G` to other protein LMs and graph SSMs in terms of enabling universal representations, AA token-level fine granularity, long-context capability, graph contextual information, large model size, and a large number of pretrained tokens.

## 2.2 LONG-CONTEXT LMS FOR BIOLOGICAL SEQUENCES

To model the long-range interactions without sacrificing single nucleotide level resolution, long-context capable LM architectures have been developed for DNA sequences, including HyenaDNA (Nguyen et al., 2024b), Evo (Nguyen et al., 2024a), and Caduceus (Schiff et al., 2024). These studies have shown that alternative architectures based on SSMs exhibit better scaling laws than Transformers on genomic data and DNA-specific tasks. Protein sequence LMs with alternative architectures have also been explored to improve computational efficiency and enable the modeling of longer protein sequences. For instance, CARP (Yang et al., 2024) is a protein LM with dilated convolution layers. Pretrained with MLM objective, CARP achieved comparable pretraining scaling properties with its Transformer counterpart ESM-1b (Rives et al., 2021) and scales better on long sequences. ProtHyena (Zhang, 2024) is a small 1.6M parameter decoder-only LM based on the Hyena operator pretrained on protein sequences and has demonstrated some improvement over ProtGPT (Ferruz et al., 2022) with comparable model sizes.

Some works exploit the long-context capability of LMs to model sets of homologous protein sequences such as those in multiple sequence alignment (MSA), which further organize a set of protein sequences by aligning evolutionary conserved amino acids across the set of sequences. PoET (Truong Jr & Bepler, 2024) proposed a tiered variant of Transformer to model the invariant relationships between multiple protein sequences from MSAs, whereas ProtMamba (Sgarbossa et al., 2024) trains a Mamba-based protein LM using concatenated sequences from MSAs with causal language modeling and infilling objective. PTM-Mamba (Peng et al., 2024) addresses post-translational modifications (PTM) of protein sequences introducing PTM tokens to amino acid tokens and subsequently trains a bidirectional Mamba model with these PTM tokens. We provide additional discussion on other pLMs and related variants in Appendix C.

Instead of training protein sequences from very specific types of data like MSAs or PTMs, we emphasize that **our work** focuses on long-context modeling of individual protein sequences and related protein sequences within biomedical graphs, which learns universal AA token-level protein

representations that are more generalizable and can encode information from biological interactions (see Table 1 for a detailed comparison).

### 2.3 PROTEIN LMS TRAINED ON GRAPHS

Graphs are ubiquitous in biomedical domains as they are suitable for organizing and representing complex biological systems, such as gene regulatory networks and PPI graphs (Wang et al., 2022). The relationships among proteins embedded in biomedical graphs have also been used to train pLMs. The common strategies for incorporating graph information include pretraining LMs with graph-specific objectives in addition to self-supervised LM objectives. The graph-specific objective can be link-prediction on homogeneous graphs (Yasunaga et al., 2022; McDermott et al., 2023), knowledge graph embedding (KGE) objectives (Zhang et al., 2022) or contrastive loss (Wang et al., 2023) on heterogeneous graphs. One limitation of such approaches is the inability to jointly model the implicit token-wise interactions beyond a pair of sequences. After all, link-prediction and KGE only take two sequences as input. In our work, we use homogeneous PPI graphs and exploit the long-context capability of SSM-based LM to model token-wise interactions beyond two sequences.

## 3 PRELIMINARIES

**Structured State Space Models**  Modern Structured SSMs are derived from first-order differential equations that map the input sequence $x(t)$ to the output sequence $y(t)$ through hidden state $h(t)$:

$$\mathbf{h}'(t) = \mathbf{A}\mathbf{h}(t) + \mathbf{B}x(t), \quad y(t) = \mathbf{C}\mathbf{h}(t) \tag{1}$$

where $\mathbf{A} \in \mathbb{R}^{N \times N}, \mathbf{B} \in \mathbb{R}^{N \times D}$ and $\mathbf{C} \in \mathbb{R}^{D \times N}$. The variables $N$ and $D$ refer to the state dimension and the (expanded) input dimension respectively. The continuous dynamical system characterized by $\mathbf{A}, \mathbf{B}$ can be discretized to $\bar{\mathbf{A}}, \bar{\mathbf{B}}$ by zero-order holding and time sampling at intervals of $\Delta$, defined as follows:

$$\bar{\mathbf{A}} = \exp(\Delta\mathbf{A}), \quad \bar{\mathbf{B}} = (\Delta\mathbf{A})^{-1}(\exp(\Delta\mathbf{A}) - \mathbf{I}) \cdot \Delta\mathbf{B}. \tag{2}$$

The formula of a discretized SSM can then be written as:

$$\mathbf{h}_k = \bar{\mathbf{A}}\mathbf{h}_{k-1} + \bar{\mathbf{B}}x_k, \quad y_k = \mathbf{C}\mathbf{h}_k \tag{3}$$

The main benefit of discretized SSMs (Gu et al., 2021) over their continuous counterpart is that they can be trained efficiently using their parallel convolutional representation and can be efficiently deployed at inference time with their recurrent form. However, the ability to model long-range interactions of SSMs is limited by the impulse response of the discrete dynamical system they implement, the S4 model (Gu et al., 2020; 2021) mitigates such limitation by introducing the HIPPO Matrix to the initialization of $\mathbf{A}$.

**Selection Mechanism and Mamba**  The main limitation of the SSMs described so far is that they cannot model complex input-varying interactions across the sequence dimension. Thus, Mamba (Gu & Dao, 2024) parameterizes the matrices $\mathbf{B}, \mathbf{C}$ and $\Delta$ in an input-dependent (data-driven) manner, introducing a selection mechanism into the S4 model. However, introducing such data dependency makes the parallelizable convolutional representation unfeasible, hence, Mamba uses a novel hardware-aware parallel computing algorithm (based on the associative scan) to ensure the efficient training of the model and leading to linear computational complexity and outstanding capabilities in modeling long-term dependencies. A Mamba model (*selective* SSM) that enables dependence of the parameters $\mathbf{B}, \mathbf{C}$ and $\Delta$ on the input $x(t)$ can be formulated as:

$$\mathbf{B}_t = \text{Linear}_{\mathbf{B}}(x_t) \quad \mathbf{C}_t = \text{Linear}_{\mathbf{C}}(x_t) \tag{4}$$

$$\Delta_t = \text{softplus}(\text{Linear}_{\Delta}(x_t)), \tag{5}$$

where $\text{Linear}(\cdot)$ represents a linear projection and $\text{softplus}(\cdot) = \log(1 + \exp(\cdot))$.

## 4 `LC-PLM`: LONG CONTEXT PROTEIN LANGUAGE MODEL

In this section, we first introduce the design choice of using bidirectional Mamba (`BiMamba`) with shared projection layers (`BiMamba-S`) for building up the model architecture of `LC-PLM`, and then

we discuss how we develop the two-stage training recipe to obtain high-quality universal protein representations using MLM and encode biologically meaningful interaction information with a novel graph context-aware training approach.

## 4.1 **BiMamba-S**: Bidirectional Mamba with Shared Projection Layers

**BiMamba** is an extension from standard Mamba block and has been applied in various domains, e.g. time-series forecasting (Liang et al., 2024), audio representation learning (Erol et al., 2024), visual representation learning (Zhu et al., 2024), DNA modeling (Schiff et al., 2024), and graph learning (Behrouz & Hashemi, 2024). The following reasons suggest we consider **BiMamba** as the design choice: (i) Mamba is good at *capturing long-range dependencies* and *extrapolating on even longer sequences*, which benefit a lot of downstream tasks on protein complexes and PPI graphs. (ii) standard Mamba only does unidirectional (associative) scans for causal modeling of sequential data. To *perform MLM to learn high-quality universal protein representations*, we introduce a modified bidirectional scan to capture information from both ends.

In general, the $l$-th **BiMamba** block takes in an input sequence of tokens $\mathbf{T}_{l-1} \in \mathbb{R}^{B \times S \times D}$ and output $\mathbf{T}_l \in \mathbb{R}^{B \times S \times D}$ where $B, S, D$ represent the batch size, the input dimension, and the hidden state dimension. Then a residual connection adds the input and output together to get $\mathbf{T}_l$. After going through $L \times$ **BiMamba** blocks, the output $\mathbf{T}_L$ will be normalized first and then fed into a prediction head to get final scores. This procedure can be formulated as follows:

$$\mathbf{T}_l = \mathrm{BiMamba}\left(\mathbf{T}_{l-1}\right) + \mathbf{T}_{l-1} \tag{6}$$

$$\hat{p} = \mathrm{PredictionHead}\left(\mathrm{Norm}\left(\mathbf{T}_L\right)\right) \tag{7}$$

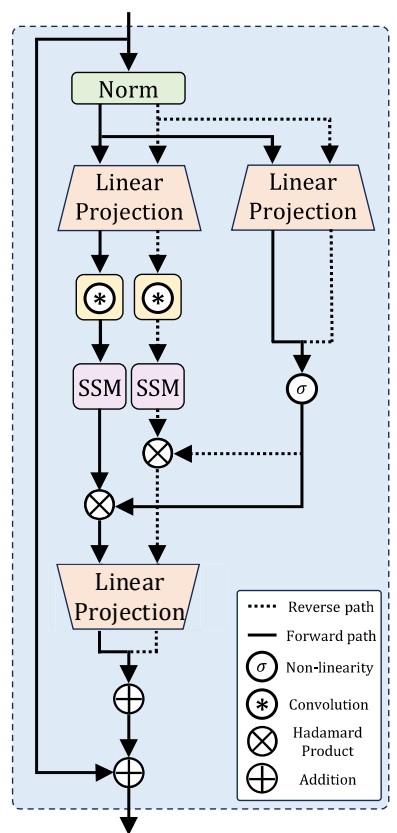

Figure 2: **BiMamba-S** block. The forward and reverse modules share the linear projection layers. The normalized input will be reversed along the sequence dimension before being fed in. The output of the reversed will be flipped back and then added to the forward's output.

Specifically, in one **BiMamba** block, the input sequence $\mathbf{T}_{l-1}$ and the flipped $\hat{\mathbf{T}}_{l-1}$ will be first normalized and then linearly projected to $\mathbf{X}_{l-1} \in \mathbb{R}^{B \times S \times E}$ and $\mathbf{Z}_{l-1} \in \mathbb{R}^{B \times S \times E}$. $\mathbf{X}_{l-1}$ and the flipped $\hat{\mathbf{X}}_{l-1}$ will be fed into the forward and inverse Mamba block respectively for a bidirectional scan. In each Mamba block, $\mathbf{X}_{l-1}$ and $\hat{\mathbf{X}}_{l-1}$ will be first passed through a 1-D convolution layer and a SiLU activation (Nwankpa et al., 2018), and then linearly projected to $\mathbf{B}_{l-1} \in \mathbb{R}^{B \times S \times N}, \mathbf{C}_{l-1} \in \mathbb{R}^{B \times S \times N}, \Delta_{l-1} \in \mathbb{R}^{B \times S \times E}$ and $\hat{\mathbf{B}}_{l-1}, \hat{\mathbf{C}}_{l-1}, \hat{\Delta}_{l-1}$, where $\Delta_{l-1}$ and $\hat{\Delta}_{l-1}$ transform $\mathbf{A}_{l-1}, \mathbf{B}_{l-1}$ and $\hat{\mathbf{A}}_{l-1}, \hat{\mathbf{B}}_{l-1}$ to $\bar{\mathbf{A}}_{l-1} \in \mathbb{R}^{B \times S \times E \times N}, \bar{\mathbf{B}}_{l-1} \in \mathbb{R}^{B \times S \times E \times N}$ and $\hat{\bar{\mathbf{A}}}_{l-1}, \hat{\bar{\mathbf{B}}}_{l-1}$. A standard SSM block will be then applied to obtain $\mathbf{Y}_{l-1} \in \mathbb{R}^{B \times S \times E}$ and $\hat{\mathbf{Y}}_{l-1}$, which later will be gated by $\mathbf{Z}_{l-1}$ and added together to get the candidate output. Lastly, a residual connection will be applied on a linear projection of the candidate output and input sequence $\mathbf{T}_{l-1}$ to get the final output $\mathbf{T}_l$. We provide an algorithmic block and a detailed and itemized procedure in Appendix I.

**Shared Projection Layers** To explore a more efficient implementation of **BiMamba**, we propose to use the shared linear projection layers for the forward input $\mathbf{T}_{l-1}$ and the flipped $\hat{\mathbf{T}}_{l-1}$. This design choice helps make the entire model $2 \times$ deeper with almost the same parameter counts (Schiff et al., 2024). We refer to this building block as **BiMamba-S** (illustrated in Figure 2). Note that this is different from the inner **BiMamba** block used in Zhang et al. (2024a); Zhu et al. (2024), where they just flipped the linearly projected hidden states. We also find that, empirically, the deeper model using **BiMamba-S** shows superiority in terms of sample & compute efficiency (4.5% improvement on evaluation loss) and performance gain on downstream tasks (an average of 4.1% improvement on TM score of structure prediction) as we expected. More results and details are shown in Section 5.3.

**Untied Input & Output Embeddings**   Notably, we opt to use untied input and output embeddings for the `BiMamba-S` encoder. Empirically we find that untied embeddings yield better evaluation loss during MLM training compared to tied embeddings, despite the latter being the standard practice. This finding aligns with previous research (Gao et al., 2019; Ethayarajh, 2019), which highlights that tying input and output embeddings leads to *anisotropic* word embeddings in contextualized pretrained models, significantly constraining their expressiveness.

### 4.2   TWO-STAGE TRAINING RECIPE

Our training procedure can be decomposed into two stages: (i) long-context protein language modeling and (ii) protein language modeling within graph context. The first stage will enforce `LC-PLM` to learn the universal token-level representations of individual proteins and the second stage will put the protein sequences into the related graph context and `LC-PLM-G` will learn to capture biologically meaningful interaction information.

**Long-context Protein Language Modeling**   `LC-PLM` is trained with `BiMamba-S` on individual protein sequences where it can leverage the power of SSM modules to effectively capture long-range dependencies within sequences. Specifically, treating the protein sequences as a collection of amino acid (AA) tokens, the model learns fine granular and universal token-level representations using MLM, which can be generalized across different types of protein sequences. For the masking strategy, we follow BERT (Devlin, 2018) in which 15% of AA tokens in a protein sequence will be "masked". Of the 'masked' tokens, 80% are replaced with [MASK], 10% are replaced with a random token from the vocabulary, and 10% are left unchanged.

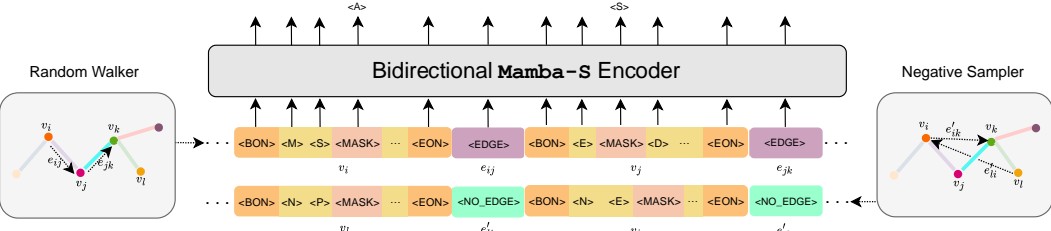

Figure 3: The illustration of graph-contextual protein language modeling (`LC-PLM-G`). The positive paths are sampled with random walks on the graph and the same number of negative paths are sampled from disconnected node pairs randomly. The sampled paths will be transformed into multi-protein sequences composed of AA tokens and special graph identifier tokens. The `BiMamba-S` encoder is trained using MLM, the same as in the first-stage training.

**Graph-contextual Protein Language Modeling**   To encode biologically meaningful interaction information into protein representations, we propose the second-stage training within a graph context where a node represents an individual protein sequence and an edge indicates the existence of PPI. We refer to this graph-contextually trained model variant as `LC-PLM-G`. Wang et al. (2022) and Behrouz & Hashemi (2024) propose to tokenize the graph into either flattened node sequences with prioritization strategy (e.g., node degree) or induced subgraphs. However, the former discards the graph topology information and the latter only provides non-Euclidean subgraph tokens that cannot be used as the input of language models. Therefore, we propose to construct the graph-contextual input via random walks (Perozzi et al., 2014; Grover & Leskovec, 2016), which can both effectively capture the graph topology and provide 1-D sequences. Consider an undirected, unweighted, PPI graph $\mathcal{G} = (V, E)$. Formally, a random walk of length $l$ can be simulated by

$$P(n_i = v \mid n_{i-1} = u) = \begin{cases} \frac{\pi_{uv}}{Z} & \text{if } (v, u) \in E \\ 0 & \text{otherwise} \end{cases} \qquad (8)$$

where $n_i$ denotes the $i$th node in the walk, $\pi_{uv}$ is the unnormalized transition probability between nodes $(u, v)$, and $Z$ is the normalizing constant. We also set two parameters $p$ and $q$ as in Grover & Leskovec (2016) to interpolate the behavior of random walker in between breath-first and depth-first search (see Appendix J). Then, the nodes in each random walk will be expanded as a sequence of

proteins composed of AA tokens. We also sample a sequence of disconnected nodes of the same length $l$ as the negative paths.

Although this gives us a principled way to form input multi-protein sequences for language models within graph context, the input still needs special identifiers to let the language model precept the graph topological information and be aware of which protein each AA token belongs to. Thus, we design four new tokens (`[BON]`, `[EON]`, `[EDGE]`, `[NO_EDGE]`) to help encode such graph context information, where the first two indicate the begin and end of a node and the last two represent if there exists an edge. We provide a visual illustration of this graph-contextual training regime in Figure 3.

## 5 EXPERIMENTS

We conduct experiments to evaluate the effectiveness of `LC-PLM` and `LC-PLM-G` and their building block `BiMamba-S`. We will address the following research questions. **(RQ1)** What is the scaling behavior of `LC-PLM`? How does it compare with its Transformer-based counterpart ESM-2? **(RQ2)** Does `LC-PLM` show stronger length extrapolation capability than ESM-2? **(RQ3)** Will `BiMamba-S` architecture be more effective in long-context protein language modeling? **(RQ4)** Does long-range dependencies help with protein structure prediction? **(RQ5)** Does `LC-PLM-G` learn graph-contextual (relational) information? **(RQ6)** Does `LC-PLM-G` with biological interaction information learned in the second-stage training help with common downstream tasks? **(RQ7)** Does `LC-PLM-G` improve protein function prediction and link prediction on the PPI graph? (See Appendix A) We provide the experimental setup, dataset description, and task definition in Appendices D and E.

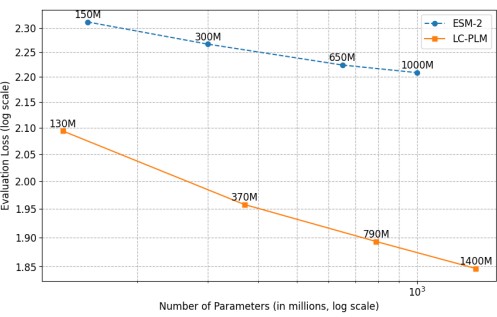
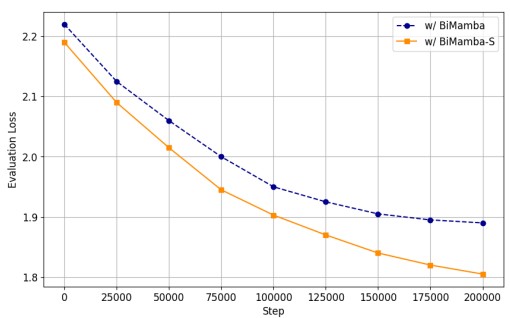

Figure 4: Evaluation loss across different model sizes for `LC-PLM` and ESM-2, showing that `LC-PLM` has a better scaling behavior when increasing parameter size.

Figure 5: Evaluation loss comparison between `LC-PLM` with `BiMamba` and `BiMamba-S` at different training steps.

### 5.1 (RQ1) EXPLORING THE SCALING LAW

We train `LC-PLM` on 20B `UniRef90` sequences and evaluate it on a held-out set of 250K `UniRef90` sequences. We test four different model sizes for both `LC-PLM` and ESM-2. The model sizes for `LC-PLM` are 130M, 370M, 790M, and 1.4B parameters to accommodate `BiMamba-S` architecture, while for ESM-2, they are 150M, 300M, 650M, and 1B. The results demonstrated that `LC-PLM` not only achieved better evaluation loss (average cross-entropy across all tokens) with a similar model size (with an average of 13.5% improvement) compared to ESM-2 but also exhibited superior scaling behavior (sharper slope) when increasing the model size, as shown in Figure 4. This aligns with the discovery in Gu & Dao (2024) that Mamba has better neural scaling law compared to Transformers in language modeling. This may also be due to the useful long-range dependencies in protein sequences captured by `LC-PLM` and the deeper architecture achieved with `BiMamba-S`.

### 5.2 (RQ2) LENGTH EXTRAPOLATION EVALUATION

We split the UniRef90 sequences into 7 bins w.r.t. the sequence length (i.e. 0-128, 128-256, 256-512, 512-1024, 1024-2048, 2048-4096, and 4096-8192). We train three sizes of `LC-PLM` (130M, 370M, 790M) and ESM-2 (150M, 300M, 650M) on the bin of 128-256 and then evaluate them on all bins (including a held-out set of 128-256). Our findings show that `LC-PLM` maintains low evaluation loss across sequence lengths, while ESM-2 struggles with both shorter and longer sequences, especially

when the lengths are underrepresented in the training set. This concludes that **LC-PLM** can extrapolate better with length due to the stronger length extrapolation capability of **BiMamba-S** (Gu & Dao, 2024) compared to ESM-2, which uses RoPE (Su et al., 2024) to extend context beyond pretraining. The results are shown in Figure 6.

### 5.3 (RQ3) The Effectiveness of **BiMamba-S**

Using shared linear projection layers in **BiMamba-S** allows for $2\times$ deeper models with similar parameter counts. In our analysis, we compare the evaluation loss of our 790M model with **BiMamba-S** and its **BiMamba** counterpart that halves the depth. The training set is UniRef50 and the evaluation set is a held-out set of 250K UniRef90 sequences, the same as in the scaling law experiments. Our results show that this parameter-efficient approach to increasing the model depth effectively improves evaluation loss by 4.5%, as shown in Figure 5. We also verify the effectiveness of **BiMamba-S** on structure prediction in Table 14, where the deeper

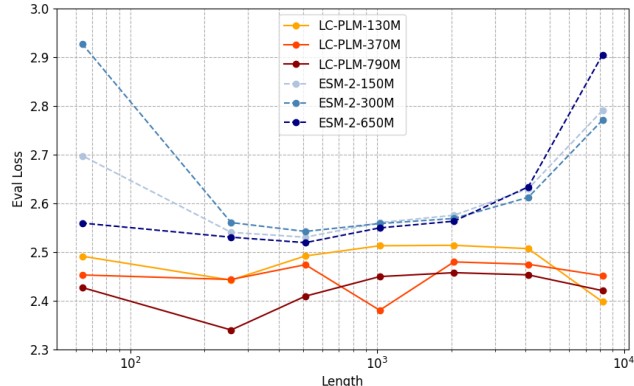

Figure 6: Length extrapolation results comparing **LC-PLM** versus ESM-2 on evaluation loss across different sequence lengths. **LC-PLM** can achieve consistent performance when extrapolating on longer sequences.

model improves by 6.7% on CASP15-multimers, 4.6% on CASP14, and 1.5% on Benchmark2. This empirical evidence matches the theory that more hidden layers in deep neural networks can benefit from more representation power gain, proposed in (Telgarsky, 2016). This also suggests a potentially better scaling strategy for training pLMs with the fixed parameter count, i.e. stacking more layers instead of having more hidden units.

### 5.4 (RQ4) Protein Structure Prediction with LMFold

We also evaluate **LC-PLM**'s ability to predict protein's 3-D structures. It has been shown that protein LMs capture various levels of protein structure information (Rives et al., 2021; Rao et al., 2020; Lin et al., 2023), despite being trained on primary sequences alone. Inspired by the ESMFold (Lin et al., 2023) architecture, which uses the residue-level embeddings and, optionally, attention maps as features to predict the 3-D structures directly without MSA[1], we developed a protein folding model named LMFold, which generalizes ESMFold's Folding Trunk and Structure Module to work with protein LMs with decoder-only and encoder-decoder architectures. Briefly, LMFold takes the residue-level embeddings for a given protein sequence as features to predict the all-atom coordinates of the protein structure. To further simplify LMFold, we only use 1 folding block of the structure module. Note that the goal of this task is not to develop the state-of-the-art protein folding model, but rather to quantify the potential of pretrained protein LMs for their learned structural information. To train LMFold, we use the *Frame Aligned Point Error (FAPE)* and *distogram* losses introduced in AlphaFold2 (Jumper et al., 2021), as well as heads for predicting *LDDT* and the *pTM score*. We weigh these 4 loss terms using the default constants proposed in OpenFold (Ahdritz et al., 2024).

| Model (#Tokens trained) | CASP15-multimers | CASP14 | Benchmark2 |
|---|---|---|---|
| ESM-2-650M (100B) | $0.4228 \pm 0.0065$ | $0.3531 \pm 0.0076$ | $0.4859 \pm 0.0119$ |
| **LC-PLM**-790M w/ **BiMamba** (100B) | $0.4787 \pm 0.0013$ | $0.3973 \pm 0.0019$ | $0.6199 \pm 0.0151$ |
| **LC-PLM**-790M w/ **BiMamba-S** (100B) | $\mathbf{0.5109 \pm 0.0070}$ | $\mathbf{0.4154 \pm 0.0080}$ | $\mathbf{0.6290 \pm 0.0071}$ |
| ProtMamba-public | N/A[2] | $0.3288 \pm 0.0091$ | $0.4515 \pm 0.0062$ |
| ESM-2-650M-public (1T)[3] | $0.5031 \pm 0.0094$ | $0.4359 \pm 0.0033$ | $0.6743 \pm 0.0067$ |

Table 2: Structure prediction performance (*TM score*) on CASP15-multimers, CASP14, and Benchmark2. We perform 3 runs using different seeds and report the mean and standard deviation.

---

[1]We disable attention maps in our experiments since (i) there is no attention map in **BiMamba-S** and (ii) ESMFold (Lin et al., 2023) also demonstrate that attention maps provide no performance gain during training.

For the training set, we down-sample 1.5% of protein chains used in OpenFold (Ahdritz et al., 2024), leading to 7,872 chains, with at most 1 protein chain from each cluster. The aggressive down-sampling is supported by the fact that training a protein folding model with as few as 1,000 protein chains achieved a decent performance (Ahdritz et al., 2024). The down-sampled protein chains have lower than 40% sequence identity to each other. We use 95% and 5% as data splitting for training and validation sets. For held-out test sets, we use CASP15-multimers (52 protein complexes), CASP14 (37 protein structures), and Benchmark2 (17 heterodimers structures) (Ghani et al., 2021). We compare our 790M **LC-PLM** (with **BiMamba** or **BiMamba-S**) against 650M ESM-2, all pretrained on 100B tokens from UniRef50. **LC-PLM** outperforms ESM-2 across all test sets by a large margin (20.8% on CASP15-multimers, 17.6% on CASP14, and 29.5% on Benchmark2). **LC-PLM** also achieves comparable performance to 650M public ESM-2 model trained on 10× more tokens (1T), with 1.6% improvement on CASP15-multimers. These results demonstrate the powerful long-context capability of **BiMamba-S** on modeling longer proteins and protein complexes. This also suggests that, even for average-length protein sequences, long-range dependencies would be useful information and an important feature for protein structure prediction.

## 5.5 (RQ5) LC-PLM-G ENCODES GRAPH RELATIONAL INFORMATION

To evaluate if **LC-PLM-G** encodes graph relational information, we first conduct the graph-contextual protein language modeling on the PPI graph provided by ogbn-proteins dataset Hu et al. (2020), which contains proteins from 8 species, to get a well-trained **LC-PLM-G**. After training, we sample the same number of proteins from each species and obtain their representations using both **LC-PLM** and **LC-PLM-G**. Since we sampled a subset of proteins (nodes) from the dataset, we can use them to construct a subgraph as well.

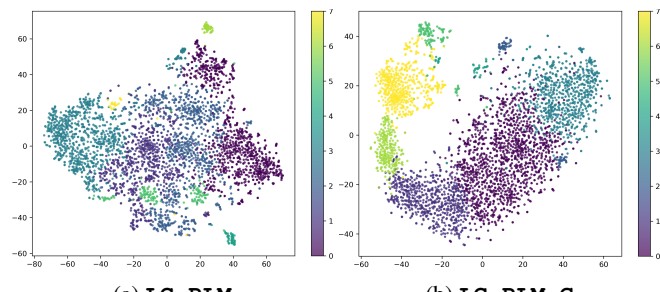

(a) **LC-PLM**      (b) **LC-PLM-G**

Figure 7: Comparison of t-SNE visualizations on protein-level representations obtained from **LC-PLM** and **LC-PLM-G** with the corresponding community labels detected by the Louvain algorithm on the PPI graph. Embeddings from **LC-PLM-G** recapitulate the topological information.

We then use the Louvain algorithm (De Meo et al., 2011) to detect 8 communities (corresponding to 8 species) in this subgraph. Next, we use t-SNE (Van der Maaten & Hinton, 2008) to reduce the dimensionality of both sets of protein embeddings obtained from **LC-PLM** and **LC-PLM-G** and label each data point using its community membership. As shown in Figure 7, the embeddings from **LC-PLM-G** captures the graph topology much better than **LC-PLM**, which aligns with the community detection results. This suggests that our proposed graph context-aware second-stage training captures the topological information in PPI graphs as expected.

## 5.6 (RQ6) INTERACTION INFORMATION HELPS DOWNSTREAM TASKS

**TAPE** Here we ask whether the graph relational information is helpful for common downstream tasks. We first use the remote homology detection and secondary structure prediction tasks from TAPE (Rao et al., 2019), which represent protein-level and residue-level tasks, respectively. Our results in Table 13 show that **LC-PLM** achieves significantly better performance in both tasks compared to ESM-2 pretrained with the same number of tokens. Remarkably, **LC-PLM** and **LC-PLM-G** even outperformed the public ESM-2 pretrained with 1T tokens, underscoring the sample efficiency of **BiMamba-S**-based model architecture can translate to downstream tasks in a supervised fine-tuning setting. The marginal improvement of **LC-PLM-G** over **LC-PLM** in remote homology tasks also suggests the information from the PPI graph helps determine protein's remote homologs, while not helpful in predicting their secondary structures at the AA level.

---

[2]ProtMamba cannot extrapolate to sequence > 2048 since they train with fixed-length positional encodings.

[3]The public ESM-2 model is provided for reference only. We highlight the best results for models trained with the same number of tokens and similar sizes. The tables below follow the same approach.

| Model (#Tokens trained) | PPI graph | Contact Map | Remote Homology | Secondary Structure |
|---|---|---|---|---|
| ESM-2-650M (100B) | None | 44.05 | $26.57 \pm 0.49$ | $79.86 \pm 0.09$ |
| ESM-2-G-650M (100B) | ogbn-proteins | 32.35 | $25.60 \pm 0.77$ | $79.76 \pm 0.24$ |
| ESM-2-G-650M (100B) | ogbl-ppa | 26.66 | $27.18 \pm 0.63$ | $79.91 \pm 0.24$ |
| **LC-PLM**-790M (100B) | None | 47.10 | $35.14 \pm 1.69$ | $\mathbf{85.07 \pm 0.03}$ |
| **LC-PLM-G**-790M (100B) | ogbn-proteins | 47.15 | $\mathbf{35.74 \pm 0.93}$ | $85.02 \pm 0.11$ |
| **LC-PLM-G**-790M (100B) | ogbl-ppa | **47.23** | $35.60 \pm 1.45$ | $85.01 \pm 0.03$ |
| ProtMamba-public | None | 10.96 | $17.82 \pm 1.85$ | $68.43 \pm 0.06$ |
| CARP-640M-public | None | 25.83 | $28.0 \pm 0.8$ | $83.0 \pm 0.1$ |
| ESM-2-650M-public (1T) | None | 66.85 | $33.43 \pm 0.35$ | $84.30 \pm 0.15$ |

Table 3: Evaluation on TAPE tasks in zero-shot (contact map) and supervised fine-tuning (remote homology and secondary structure) settings. We report the *Precision@2/L* for Contact Map prediction, *top-1 accuracy* for the Remote Homology fold-level test set, and *accuracy* for the 3-class secondary structure prediction on the CB513 test set, respectively. Values for CARP are taken from Yang et al. (2024). We perform 3 runs using different seeds to report the mean and standard deviation.

**ProteinGym**   Next, we evaluate the predicted fitness landscape on zero-shot mutation effect prediction. It has been shown that pretrained pLMs can capture the fitness landscape of proteins without any further training (Meier et al., 2021). We use 217 deep mutational scan (DMS) datasets collected in ProteinGym (Notin et al., 2024), which collectively measure the effects of 2.5 million substitution mutations to parent protein sequences. In Table 4, we demonstrate **LC-PLM** achieved significantly better alignment with protein fitness compared to ESM-2 pretrained with the same number of tokens. Interestingly, we note that the graph-contextual training hurts the fitness landscapes of ESM-2 models, while **LC-PLM-G** retained their zero-shot capabilities for protein fitness prediction. We hypothesize that the long graph context degrades the representation space of ESM-2, not for **LC-PLM-G**. This highlights **BiMamba-S** as a superior architectural design choice, demonstrating robustness in maintaining performance across various tasks while excelling in those that benefit from interaction information learned through graph-contextualized training, possibly due to its preferable context-length extrapolation property.

| Model (#Tokens trained) | PPI graph | Spearman | NDCG |
|---|---|---|---|
| ESM-2-650M (100B) | None | $0.295 \pm 0.013$ | $0.695 \pm 0.008$ |
| ESM-2-G-650M (100B) | ogbn-proteins | $0.109 \pm 0.013$ | $0.642 \pm 0.008$ |
| ESM-2-G-650M (100B) | ogbl-ppa | $0.131 \pm 0.014$ | $0.644 \pm 0.007$ |
| **LC-PLM**-790M (100B) | None | $0.378 \pm 0.008$ | $\mathbf{0.735 \pm 0.005}$ |
| **LC-PLM-G**-790M (100B) | ogbn-proteins | $\mathbf{0.380 \pm 0.008}$ | $0.734 \pm 0.006$ |
| **LC-PLM-G**-790M (100B) | ogbl-ppa | $\mathbf{0.380 \pm 0.008}$ | $0.734 \pm 0.006$ |
| ESM-2-650M-public (1T) | None | $0.414 \pm 0.011$ | $0.747 \pm 0.005$ |
| ESM-2-3B-public (1T) | None | $0.406 \pm 0.011$ | $0.755 \pm 0.004$ |
| PoET (Truong Jr & Bepler, 2023) | None | 0.479 | N/A |
| TranceptEVE-L (Notin et al., 2022) | None | 0.454 | 0.786 |
| SaProt (Su et al., 2023) | None | 0.457 | 0.768 |

Table 4: Evaluation on ProteinGym DMS substitutions benchmark. We report *Spearman's correlation coefficient* and *normalized discounted cumulative gain (NDCG)* between the log odds ratio and the experimentally measured protein fitness scores for each DMS assay.

## 6   CONCLUSION

In this work, we explored **LC-PLM** and **LC-PLM-G** based on **BiMamba-S**. We demonstrate **LC-PLM**'s favorable neural scaling laws and length extrapolation property than Transformer-based ESM-2. We found that this length extrapolation property can facilitate the 3-D structure prediction of longer proteins and protein complexes. Specifically, **LC-PLM** achieved 7% to 34% better performance on various downstream tasks. We also found that after training within graph context using random walk sampling, **LC-PLM-G** can capture relational structure encoded in protein-protein interactions and improve remote homology prediction by more than 35% compared to ESM-2.

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

CONTENTS

# Appendices

## APPENDIX A  (**RQ7**) PROTEIN FUNCTION PREDICTION AND LINK PREDICTION ON PPI GRAPH

We evaluate `LC-PLM-G` on two tasks: protein function prediction (`ogbn-proteins`) and PPI link prediction (`ogbl-ppa`). On `ogbn-proteins`, `LC-PLM-G` achieves an *accuracy* of 0.8925 $\pm$ 0.001, outperforming the state-of-the-art by 2.6%. For `ogbl-ppa`, we leverage the learned embeddings from both `LC-PLM` and `LC-PLM-G` to initialize the node attributes for GCN and GraphSAGE, evaluating these model variants. The results confirm that the embeddings improve performance. We conduct similar experiments on `ogbn-proteins`, further validating the effectiveness of capturing graph-contextual information. Additional details are provided in Appendix H.

## APPENDIX B  DISCUSSION, AND FUTURE WORK

In this work, we explored `LC-PLM` and `LC-PLM-G` based on `BiMamba-S`. `BiMamba-S` can be formulated in a more theoretical way as structured SSMs with quasi-separable matrices (Hwang et al., 2024a). We did not apply quasi-separable mixers in our model since Mamba currently has much better software-hardware interface and distributed training support in practical implementations that help train large foundation models feasibly and efficiently. We demonstrate `LC-PLM`'s favorable neural scaling laws and length extrapolation property than Transformer-based ESM-2. We found that this length extrapolation property can facilitate the 3-D structure prediction of longer proteins and protein complexes. Specifically, `LC-PLM` achieved 7% to 34% better performance on various downstream tasks. We also found that after training within graph context using random walk sampling, `LC-PLM-G` can capture relational structure encoded in protein-protein interactions and improve remote homology prediction by more than 35% compared to ESM-2.

`LC-PLM` not only demonstrates superior performance on longer protein sequences but also outperforms ESM-2 on shorter protein sequences, highlighting a significant performance gap between SSMs and Transformers in protein language modeling. We hypothesize that this advantage may be attributed to the relatively small vocabulary size of protein sequences ($\sim$ 20 tokens), which allows SSMs to more effectively learn compressed state representations compared to natural languages, where the vocabulary size typically exceeds 50k tokens.

It is also possible that ESM-2 could incorporate more advanced architectural designs within its Transformer-based framework to narrow the performance gap, given the rapid advancements in text modeling and Transformer architectures. There are also emerging techniques that may further assist ESM-2 in bridging the gap in long-context modeling capability compared to `LC-PLM`.

In future work, we aim to (i) explore hybrid architectures that integrate multi-head attention with SSMs, (ii) investigate more advanced self-supervised training strategies to enhance the incorporation of graph-contextual information during the later stages of pre-training, e.g., contrastive learning using *(positive, negative)* pairs of random walk paths to reinforce locality relationships, and (iii) develop more principled approaches for negative sampling to better contrast with *positive* random walk paths. Some other directions include (i) approximate permutation-invariant graph context learning for pLMs, e.g. using permutation group (Huang et al., 2022), (ii) explore other graph context extraction methods instead of random walk, e.g. graph skeleton tree (Huang et al., 2023).

We also think it deserves to expand the application scope of `LC-PLM` across several key areas: (i) understanding viral protein sequences, which are characterized by extended sequence lengths; (ii) enhancement of protein co-regulation and functional prediction capabilities (Hwang et al., 2024b); and (iii) advanced protein design tasks requiring expanded contextual understanding, such as protein inpainting. We anticipate that `LC-PLM`'s capabilities will enable novel applications beyond these identified domains, presenting significant opportunities for further exploration in the field of protein modeling and design.

## APPENDIX C   MORE DISCUSSION ON GENERAL PLMS

As noted earlier, protein sequences, represented as strings of amino acid letters, are well-suited to LMs that can capture complex dependencies among amino acids (Ofer et al., 2021a). pLMs (Hu et al., 2022) have emerged as promising tools for learning protein sequences. This section introduces LSTM-based pLMs, followed by Transformer-based pLMs, detailing their implementation strategies and applications, particularly for protein structure prediction.

Klausen et al. (2018) developed a combination of convolutional and LSTM neural networks to predict various protein structural features, such as solvent accessibility, secondary structure, structural disorder, and torsion angles ($\varphi, \psi$) for each residue. Models like SPIDER3-Single (Heffernan et al., 2018) focus on single sequences rather than relying on multiple sequence alignments (MSAs). Similarly, models such as DeepPrime2Sec (Asgari et al., 2019) and SPOT-1D-Single (Singh et al., 2021a) share comparable training objectives and architectures. Furthermore, models like DeepBLAST (Morton et al., 2020), SPOT-1D-LM (Singh et al., 2021c), and SPOT-Contact-Single (Singh et al., 2021b) utilize embeddings from pre-trained pLMs for downstream tasks such as contact map and function prediction.

However, the TAPE benchmark (Rao et al., 2019) highlighted opportunities for innovative design and training methods beyond traditional LSTMs and Transformers. UniRep (Alley et al., 2019), for example, employs a multiplicative LSTM (mLSTM)(Krause et al., 2016) to condense arbitrary protein sequences into fixed-length vectors, capturing long-range dependencies. Similarly, UDSM-Prot(Strodthoff et al., 2020) and SeqVec (Heinzinger et al., 2019) utilize LSTM variants to develop rich, transferable representations. ProSE (Bepler & Berger, 2021) enhances these representations with structural supervision through residue-residue contact loss and structural similarity prediction, while CPCProt (Lu et al., 2020) leverages InfoNCE loss to maximize mutual information in protein embeddings.

ProtTrans (Elnaggar et al., 2021) trained extensive models (including T5, ELECTRA, ALBERT, XLNet, BERT, and Transformer-XL) on sequences comprising 393 billion amino acids across 5616 GPUs and one TPU Pod. ESM-1b (Rives et al., 2019) demonstrates how deep Transformers, coupled with a masking strategy, can build intricate context-aware representations. The results from ProtTrans and ESM-1b suggest that large-scale pLMs can effectively learn the grammar of proteins, even without evolutionary data. Furthermore, PMLM (He et al., 2022) enhances model performance on the TAPE contact benchmark by accounting for dependencies among masked tokens, indicative of inter-residue coevolution.

Incorporating additional data such as MSAs, functions, structures, and biological priors can enrich protein embeddings. For instance, the MSA Transformer (Rao et al., 2021) adapts Transformer LMs to handle sets of sequences, utilizing alternating attention mechanisms. ProteinBERT (Ofer et al., 2021b) integrates sequence information with Gene Ontology (GO) annotations to predict diverse protein functions, while OntoProtein (Zhang et al., 2022) leverages GO as a factual knowledge graph. The PEER benchmarks (Xu et al., 2022) demonstrate the importance of selecting suitable auxiliary tasks to enhance model performance across a variety of protein-related tasks.

**Protein Structure Prediction**    Early pLMs (Klausen et al., 2018; Heffernan et al., 2018; Asgari et al., 2019) primarily predicted structural features, which are essential for constructing 3D protein structures. Recent models aim to predict protein structures end-to-end. Evoformer, a core module in the AF2 network (Jumper et al., 2021), exemplifies this with its sophisticated design that includes axial attention and updates to pair representations ensuring consistency principles like the triangle inequality.

**Other Applications**    ProGen (Madani et al., 2020) exemplifies models trained on sequences conditioned on specific protein properties. In contrast, newer models like ProGen2 (Nijkamp et al., 2022) and AminoBERT (Chowdhury et al., 2022) illustrate the expansion of pLM applications to include tasks like antibody structure prediction, demonstrating the versatile utility of pLMs across a range of biological research and clinical applications.

## APPENDIX D  DATASETS, TASKS, AND METRICS

### D.1  PROTEIN SEQUENCE DATASETS

We first describe the `Unified Reference Protein (UniRef)` dataset(Suzek et al., 2015), which provides clustered sets of protein sequences from the UniProt Knowledgebase (UniProtKB) (Boutet et al., 2016) and selected UniParc records. It's designed to speed up protein sequence analysis by reducing the redundancy of sequences at different levels without losing the coverage of sequence space. Here are the key features of the `UniRef` dataset:

- `UniRef100:` This dataset includes all the protein sequences from UniProtKB and selected UniParc records, clustered by exact sequence matches. It provides comprehensive coverage and serves as the basis for the other two datasets.

- `UniRef90:` This set clusters sequences that have at least 90% sequence identity and 80% overlap in alignment, compressing the dataset while still preserving most of the sequence diversity. It is used for high-throughput and large-scale analysis where a balance between speed and coverage is needed.

- `UniRef50:` This dataset clusters sequences with at least 50% sequence identity and 80% overlap in alignment, further reducing the dataset size and redundancy. It's intended for rapid scans and for exploring broad phylogenetic relationships.

Each entry in a UniRef dataset represents a cluster and contains the sequence of the representative protein (the longest sequence or the one with the most annotations), along with a list of all the cluster members. These datasets are useful for various bioinformatics tasks such as sequence alignment, phylogenetic analysis, and functional annotation, as they allow researchers to handle large volumes of sequence data more efficiently.

We use the 2024-01 release of UniRef[4], and preprocessed by removing de-novo designed proteins by filtering out protein sequences annotated by `Tax=synthetic construct`. Next, we randomly sample 250,000 sequences from `UniRef90` as the validation set to report evaluation losses for pretraining protein language models (pLMs). To remove sequences from training sets (`UniRef50` and `UniRef90`) that are highly similar to the validation set, we use the training sets as query databases and validation set as a target database by mmseqs2 (Steinegger & Söding, 2017) with the following command: `mmseqs search -min-seq-id 0.5 -alignment-mode 3 -max-seqs 300 -s 7 -c 0.8 -cov-mode 0`.

For the first-stage training of **LC-PLM** and ESM-2 models, we use the `UniRef50` training set; and for the scaling law and length extrapolation experiments, we use the `UniRef90` set. We also provide the histogram of `UniRef90` in terms of sequence length in Figure 8. The training and evaluation sets are randomly sampled from the entire set, which should follow the same distribution. The average lengths of `UniRef90` and `UniRef50` are shown in Table 5

| Database | Tokens | Num sequences | Average length |
|---|---|---|---|
| UniParc | 221.7B | 577.8M | 383.6bp |
| UniRef100 | 144.3B | 376.6M | 383.2bp |
| UniRef90 | 61.2B | 179.5M | 341.2bp |
| UniRef50 | 17.8B | 62.8M | 284.4bp |

Table 5: Average length of `UniRef`.

### D.2  STRUCTURE PREDICTION DATASETS

`CASP15-multimers` is a subset derived from the 15th edition of Critical Assessment of Protein Structure Prediction (CASP) challenge (Kryshtafovych et al., 2023), specifically focusing on predicting the structures of protein complexes or multimers. In `CASP15`, the multimer track evaluates

---

[4]https://ftp.uniprot.org/pub/databases/uniprot/previous_releases/release-2024_01/uniref/

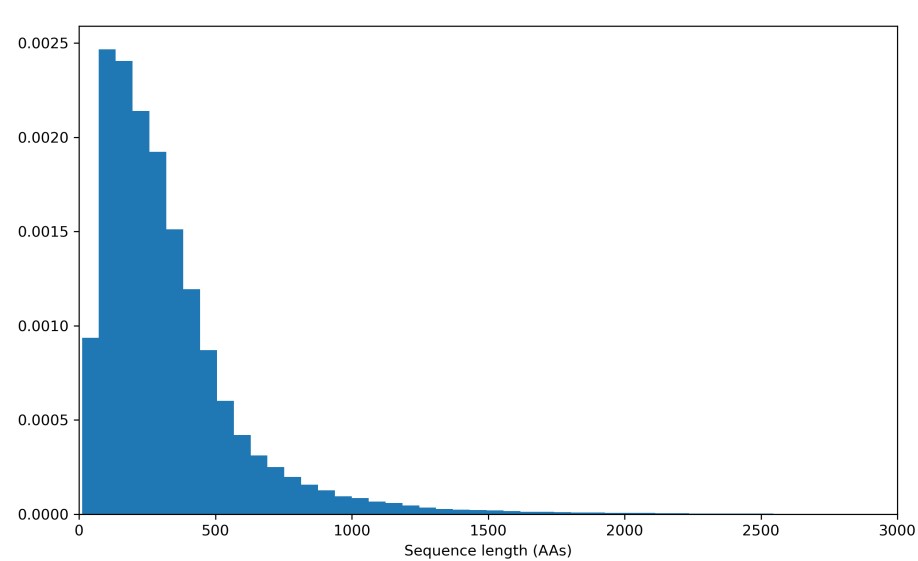

Figure 8: The sequence length distribution of `UniRef90`.

the ability of protein structure prediction methods to accurately model the quaternary structures of protein assemblies. `CASP15-multimers` includes 52 protein complexes.

`CASP14` is a dataset from the 14th edition of the CASP challenge (Kryshtafovych et al., 2021). CASP is a biennial experiment that assesses the state-of-the-art methods for protein structure prediction. `CASP14` covers a wide range of protein structure prediction tasks, including free modeling (FM), template-based modeling (TBM), and the prediction of protein domains with challenging folds. `CASP14` includes 37 protein structures.

`Benchmark2` (Ghani et al., 2021) is a dataset commonly used in the field of protein-protein docking and structure prediction. It is a curated collection of protein complexes that have been used extensively to evaluate the performance of computational methods in predicting the binding orientation of protein complexes. The dataset includes both bound and unbound forms of protein structures, providing a challenging testbed for docking algorithms. `Benchmark2` includes 17 heterodimers structures.

**Metrics** (i) `Frame Aligned Point Error (FAPE)` measures the error in aligning one set of points (predicted) to another (target), considering both translational and rotational components. The error is calculated on a per-point basis between corresponding points in the two sets, after aligning them using a frame of reference. The error is normalized by the size of the objects involved, which allows it to be invariant to the absolute size of the objects, making it suitable for tasks involving objects of varying scales. The loss is usually computed as an L1 or L2 norm of the differences in the aligned coordinates, which provides a straightforward gradient for optimization. (ii) `Distogram Loss` aims to improve the separability of feature distributions for different classes. It works by encouraging the histograms (or distributions) of distances within classes (positive pairs) to be distinct from histograms of distances between classes (negative pairs). The loss function is designed to increase the overlap between histograms of positive pairs while decreasing the overlap for negative pairs. This is achieved by calculating the probability of a randomly chosen positive pair having a smaller distance than a randomly chosen negative pair. Typically, a differentiable approximation of the histogram is used, and the optimization focuses on adjusting the model parameters to achieve the desired separation in the histograms' distributions. (iii) `pTM score` is used to assess the structural similarity between two proteins, normalized for protein size. TM scores range from 0 to 1, where a score higher than 0.5 generally indicates a model of correct topology and a score below 0.17 suggests random similarities. TM-score is more sensitive to the global fold of a protein than to the specific atomic positions, making it particularly useful in assessing larger, domain-level accuracies. (iv) `LDDT`

is a local superposition-free score that evaluates the local accuracy of a protein model by comparing distances between all atom pairs within defined cutoffs in both the predicted and reference models. It can provide a more detailed view of the quality of a pLM at the residue level.

### D.3 TAPE AND PROTEINGYM

**Tasks Assessing Protein Embeddings (TAPE)**  TAPE (Rao et al., 2019) is a set of five biologically relevant semi-supervised learning tasks spread across different domains of protein biology. We adopted four tasks: 1) Secondary Structure prediction, and 2) Remote Homology Detection, 3) Stability prediction, 4) Fluorescence prediction. Secondary structure prediction is a sequence-to-sequence task where each input amino acid is mapped to one of the three labels from {Helix, Strand, Other}. It probes the model's ability to learn local structure. Remote Homology Detection is a sequence classification task where each input protein is mapped to a label $\{1, \cdots, 1195\}$, representing different possible protein folds. This task measures models' capability to detect structural similarity across distantly related proteins. Stability and Fluorescence predictions are sequence-level regression tasks that probe the model's ability to predict the mutant sequences' properties, which are the thermostability and the flurescent intensity, respectively.

**ProteinGym**  ProteinGym (Notin et al., 2024) is collection of benchmarks aiming at comparing the ability of models to predict the effects of protein mutations. We use the DMS Substitution subset from ProteinGym, which covers 2.5 million mutants from across 217 assays.

### D.4 DATASETS FOR PROTEIN FUNCTION PREDICTION AND LINK PREDICTION ON PPI GRAPH

**ogbn-proteins**  The `ogbn-proteins` dataset is structured as an undirected, weighted graph with nodes and edges categorized by species. Nodes in this graph represent proteins, while the edges denote various biologically significant associations such as physical interactions, co-expression, or homology (Szklarczyk et al., 2019). Each edge is associated with an 8-dimensional feature vector, where each dimension quantifies the confidence level of a particular association type on a scale from 0 to 1—with higher values indicating greater confidence. The dataset includes proteins from eight different species. The objective is to predict protein functions using a multi-label binary classification approach, where there are 112 different functions to predict. The performance metric used is the average ROC-AUC score across these 112 prediction tasks. The dataset is divided into training, validation, and test sets based on the species of the proteins. This split strategy is designed to assess how well models can generalize across different species. Notably, the `ogbn-proteins` dataset lacks specific input features for nodes but includes features for over 30 million edges. In baseline experiments, a simple approach is taken where the node features are derived by averaging the features of incoming edges to each node.

**ogbl-ppa**  The `ogbl-ppa` dataset is an undirected, unweighted graph where nodes represent proteins from 58 different species, and edges illustrate biologically significant relationships between proteins, including physical interactions, co-expression, homology, or genomic neighborhood (Szklarczyk et al., 2019). Each node is described by a 58-dimensional one-hot vector indicating the species of the protein. The objective is to predict potential new association edges based on the provided training edges. The model's performance is evaluated on its ability to prioritize positive test edges over negative ones. Each positive edge in the validation or test set is ranked against 3,000,000 randomly selected negative edges. The effectiveness of the model is measured using the Hits@K metric, where K = 100 is determined to be an effective threshold in initial experiments. This metric, which requires the model to consistently rank positive edges above a vast majority of negative edges, poses a greater challenge than the ROC-AUC metric. The dataset divides edges into training, validation, and test sets based on the method used to determine the associations. Training edges consist of associations identified either through high-throughput methods (such as automated, large-scale experiments) or computationally (e.g., through text mining). Conversely, validation and test edges are derived from protein associations verified through low-throughput, labor-intensive experiments in the lab (Macarron et al., 2011; Bajorath, 2002; Younger et al., 2017). The primary challenge is to predict specific types of protein associations, like physical protein-protein interactions, based on other more readily measurable types of associations that are correlated with the target interactions.

| Name | #Nodes | #Edges | Split | Task | Metric |
|------|--------|--------|-------|------|--------|
| ogbn-proteins | 132,534 | 39,561,252 | Species | Binary classification | ROC-AUC |
| ogbl-ppa | 576,289 | 30,326,273 | Throughput | Link prediction | Hits@100 |

Table 6: Summary of OGB datasets.

# APPENDIX E    EXPERIMENTAL SETUP

## E.1    HARDWARE AND SOFTWARE

All experiments are run on NVIDIA A100 Tensor Core GPU except `ogbn-proteins` and `ogbl-ppa`, which are run on NVIDIA A10G Tensor Core GPUs. For core software packages in main experiments, we use Python 3.10, PyTorch 2.1.0, Transformers 4.41.2, DeepSpeed 0.14.4, Accelerate 0.27.2, mamba-ssm 2.2.0, datasets 2.20.0, Triton 2.0.0, and CUDA Toolkit 12.1. For some downstream tasks, the dependencies and package version will be adjusted accordingly. For `ogbn-proteins` and `ogbl-ppa`, we add several new packages: PyTorch Geometric 2.5.3, torch-cluster 1.6.3, torch-scatter 2.1.2, torch-sparse 0.6.18, torch-spline-conv 1.2.2, and OGB 1.3.6.

## E.2    MASKED LANGUAGE MODELING

The input to the model consists of raw protein sequences, which are tokenized into individual amino acids. A subset of these amino acids is randomly selected and masked during training. Typically, 15% of the amino acids in a sequence are selected for masking. Of these selected tokens, 80% are replaced with a special [MASK] token, 10% are replaced with a random amino acid, and the remaining 10% are left unchanged. The model is trained to predict the identity of these masked amino acids using a cross-entropy loss.

The training process is conducted using the AdamW optimizer with a learning rate that is linearly warmed up for a small percentage of the total training steps, followed by a cosine decay schedule. The batch size and learning rate are chosen based on the model size and computational resources, with a total number of tokens approximately equal to 0.5M[5] and learning rates are set as $2 \times 10^{-4}$. Gradient clipping is often applied to stabilize the training. Additionally, a 0.1 weight decay is applied.

For second-phase training with graph context, we sample random walks with the context length $l = 5$ on the given PPI graph (`ogbn-proteins` or `ogbl-ppa`) and retrieve the corresponding protein sequence for each node from String database (Szklarczyk et al., 2019). We then add 4 special tokens to indicate the begin and end of the node, the edge and non-edge in the random walk path. We continue MLM training the 790M **LC-PLM** on 20B more tokens on random walks to get **LC-PLM-G**, during which we freeze the parameters in SSMs and linear projection layers except the input & output embeddings and normalization layers. We summarize the key hyperparameters in table 7.

## E.3    STRUCTURE PREDICTION WITH LMFOLD

To train LMFold, we use the FAPE and distogram losses introduced in AlphaFold2 (Jumper et al., 2021), as well as heads for predicting LDDT and the pTM score. We weigh these 4 loss terms using the default constants proposed in OpenFold (Ahdritz et al., 2024). We used Adam with $\beta_1 = 0.9$, $\beta_2 = 0.99$, and $\epsilon = 1^{-6}$ as optimizers, and warmed up the learning rate linearly over the first 1,000 iterations from 0 to $1^{-3}$. We use a per-GPU batch size of 1 training on 32 NVIDIA A100 GPUs, leading to a global batch size of 32.

## E.4    EVALUATION ON TAPE TASKS

To evaluate pretrained pLMs on TAPE Remote Homology prediction and Secondary Structure prediction, we followed the evaluation setting from Rao et al. (2019). For the Remote Homology task, we add a two-layered MLP (with 512 as the intermediate dimension) on top of the protein-level

---

[5]For 130M, 370M, 790M, and 1.4B of **LC-PLM**, we train on 16, 32, 64, 128 A100s, respectively. The local batch size and the gradient accumulation steps are dynamically adjusted to ensure the model fits in.

| Hyperparameters | 130M | 370M | 790M | 1.4B |
|---|---|---|---|---|
| Peak learning rate | | $2 \times 10^{-4}$ | | |
| Global batch size | | 0.5M tokens | | |
| Block size | | 1024 | | |
| Warm-up steps | | 2000 | | |
| Adam betas | | $\beta_1 = 0.9, \beta_2 = 0.95$ | | |
| Maximum gradient norm | | 0.5 | | |
| Precision | | BF16 | | |
| Optimizer | | AdamW | | |
| Learning rate scheduler | | cosine | | |
| Weight decay | | 0.1 | | |
| Length of random walks | | 5 | | |
| Hidden size | 768 | 1024 | 1536 | 2048 |
| Number of `BiMamba-S` blocks | 24 | 48 | 48 | 48 |

Table 7: Summary of MLM training hyperparameters for `LC-PLM` and `LC-PLM-G`.

| Hyperparameters | 150M | 300M | 650M | 1.0B |
|---|---|---|---|---|
| Peak learning rate | | $2 \times 10^{-4}$ | | |
| Global batch size | | 0.5M tokens | | |
| Warm-up steps | | 2000 | | |
| Adam betas | | $\beta_1 = 0.9, \beta_2 = 0.98$ | | |
| Maximum gradient norm | | 1.0 | | |
| Precision | | BF16 | | |
| Optimizer | | AdamW | | |
| Learning rate scheduler | | cosine | | |
| Weight decay | | 0.01 | | |
| Length of random walks | | 5 | | |
| Hidden size | 640 | 960 | 1280 | 1280 |
| Intermediate size | 2560 | 3840 | 5120 | 7680 |
| Number of hidden layers | 30 | 30 | 33 | 33 |

Table 8: Summary of MLM training hyperparameters for ESM-2.

embeddings from a pLM. The protein-level embeddings are calculated as the average of token-level embeddings. For the Secondary Structure prediction task, we used a single-layered MLP taking the token-level embeddings from the LM directly to make a token-level 3-class classification.

Then, we fine-tune the parameters of the LM and the MLP prediction head end-to-end on the training set and perform early stopping on the validation set. We report the top-1 accuracy on the fold-level hold-out test set for Remote Homology task, and accuracy on the CB513 test set for the Secondary Structure prediction task, respectively. The detailed hyperparameters we used are listed in Table 9.

### E.5 EVALUATING ON PROTEINGYM

To evaluate pretrained pLMs on the ProteinGym DMS Substitution benchmarks, we adopt the masked-marginals heuristic (Meier et al., 2021) to predict protein fitness in a zero-shot setting. The masked-marginals method scores mutations (mt) using the log odds ratio at the mutated position over wild-type (wt), assuming an additive model when multiple mutations $T$ exist in the same protein sequence:

$$\sum_{t \in T} \log p(x_t = x_t^{mt} | x_{\setminus T}) - \log p(x_t = x_t^{wt} | x_{\setminus T})$$

We then compute Spearman's correlation coefficient and normalized discounted cumulative gain (NDCG) between the log odds ratio and the experimentally measured protein fitness scores for each

| Hyperparameters | Remote Homology | Secondary Structure | Stability | Fluorescence |
|---|---|---|---|---|
| Batch size | | 16 | 512 | 128 |
| Number of warm-up steps | | 5000 | | |
| Early stopping patience | | 25 epochs | | |
| Max number of epochs | | 100 | | |
| Learning rate decay schedule | | cosine | | |
| Optimizer | | AdamW | | |
| Adam betas | | $\beta_1 = 0.9, \beta_2 = 0.98$ | | |
| Peak learning rate | 1e-5 | 5e-5 | 1e-4 | 5e-5 |
| Prediction head | 2-layered MLP | 1-layered MLP | 2-layered MLP | |

Table 9: Hyperparameters used for fine-tuning protein language models for TAPE tasks.

DMS assay. The Spearman's correlation coefficients are aggregated across 217 DMS assays using the provided code.

### E.6 PROTEIN FUNCTION PREDICTION AND PPI LINK PREDICTION

For protein function prediction on `ogbn-proteins`, we use GIPA (Li et al., 2023) as the graph neural network (GNN) backbone with our learned protein sequence embeddings as the node attributes initialization. We report the hyperparameters (HPs) we use to train the model in Table 10. Furthermore, we conduct more ablation studies on the GNN backbone to test if the contribution of learned protein embeddings is independent of the architecture design choice. We choose two popular GNNs, i.e. GCN (Kipf & Welling, 2017) and GraphSAGE (Hamilton et al., 2017) to evaluate the embeddings. The HPs are shown in Table 11.

For PPI link prediction on `ogbl-ppa`, we use a combined backbone of NGNN (Song et al., 2021) and SEAL (Zhang & Chen, 2018) with labeling tricks (Zhang et al., 2021). We summarize the HPs used in this backbone in Table 12. Note that there is a ratio $k$ used in SortPooling (Zhang et al., 2018). We also perform similar ablation studies on this task using GCN and GraphSAGE with the same set of HPs reported in Table 11.

| Hyperparameters | Value |
|---|---|
| # of epochs | 1500 |
| # of heads | 20 |
| Learning rate | 0.01 |
| # of layers | 6 |
| # of hidden units | 50 |
| Dropout rate | 0.4 |

Table 10: HPs of GIPA backbone.

| Hyperparameters | Value |
|---|---|
| # of epochs | 1000 |
| # of heads | 20 |
| Learning rate | 0.01 |
| # of layers | 3 |
| # of hidden units | 256 |
| Dropout rate | 0.3 |

Table 11: HPs of GCN/SAGE.

| Hyperparameters | Value |
|---|---|
| # of epochs | 50 |
| $k$ of SortPooling | 0.6 |
| Learning rate | 0.00015 |
| # of layers | 3 |
| # of hidden units | 48 |
| Dropout rate | 0.0 |

Table 12: HPs of NGNN/SEAL.

## APPENDIX F  MORE RESULTS ON TAPE TASKS

For the Jacobian contact map prediction task, we adopted the methods from Zhang et al. (2024b) to use categorical Jacobian matrices computed from protein language models as the zero-shot prediction for protein contact maps and report the precision@2/L (L is the length of a protein sequence) on the validation set of ProteinNet dataset (AlQuraishi, 2019).

## APPENDIX G  ROBUSTNESS OF DOWNSAMPLING FOR STRUCTURE PREDICTION TRAINING SET

We perform three 1.5% downsampling using three random seeds and retrain both our LC-PLM and ESM-2. We find that the downsampling is very robust to the performance with a small standard

| Model (#Tokens trained) | PPI graph | Contact Map | Stability | Fluorescence |
|---|---|---|---|---|
| ESM-2-650M (100B) | None | 44.05 | $0.763 \pm 0.008$ | $0.695 \pm 0.002$ |
| ESM-2-G-650M (100B) | ogbn-proteins | 32.35 | $0.750 \pm 0.016$ | $0.694 \pm 0.002$ |
| ESM-2-G-650M (100B) | ogbl-ppa | 26.66 | $0.753 \pm 0.009$ | $0.693 \pm 0.001$ |
| `LC-PLM`-790M (100B) | None | 47.10 | $0.794 \pm 0.003$ | $0.692 \pm 0.002$ |
| `LC-PLM-G`-790M (100B) | ogbn-proteins | 47.15 | $0.801 \pm 0.001$ | $0.709 \pm 0.003$ |
| `LC-PLM-G`-790M (100B) | ogbl-ppa | 47.23 | $0.801 \pm 0.001$ | $0.693 \pm 0.002$ |
| ProtMamba-public | None | 10.96 | $0.726 \pm 0.012$ | $0.688 \pm 0.005$ |
| CARP-640M-public | None | 25.83 | $0.720 \pm 0.010$ | $0.680 \pm 0.002$ |
| ESM-2-650M-public (1T) | None | 66.85 | $0.804 \pm 0.006$ | $0.688 \pm 0.001$ |

Table 13: Evaluation on TAPE tasks in supervised fine-tuning setting. We report the Spearman's correlation coefficients for the test sets for the Stability and Fluorescence prediction tasks. We perform 3 runs using different seeds to report the mean and standard deviation.

deviation that is close to (even smaller than) the standard deviation of using the same train set but training with different random seeds as we reported in Table 14.

| Model (#Tokens trained) | CASP15-multimers | CASP14 | Benchmark2 |
|---|---|---|---|
| ESM-2-650M (100B) | $0.4132 \pm 0.0065$ | $0.3437 \pm 0.0039$ | $0.4773 \pm 0.0092$ |
| `LC-PLM`-790M (100B) | $0.5004 \pm 0.0139$ | $0.4244 \pm 0.0053$ | $0.6290 \pm 0.0121$ |
| ESM-2-650M-public (1T) | $0.5128 \pm 0.0003$ | $0.4421 \pm 0.0023$ | $0.6844 \pm 0.0059$ |

Table 14: Structure prediction performance (*TM score*) on `CASP15-multimers`, `CASP14`, and `Benchmark2`. We perform 3 downsamplings using different seeds and report the mean and standard deviation.

## APPENDIX H    MORE DETAILS FOR PROTEIN FUNCTION PREDICTION AND PPI LINK PREDICTION

We evaluate `LC-PLM-G`'s performance on two graph-related downstream tasks, protein structure prediction and PPI link prediction on OGB (Hu et al., 2020), i.e. ogbn-proteins and ogbl-ppa. Both datasets provide a PPI graph but with different scales (i.e. the numbers of nodes and edges) and predict different tasks (i.e. node property prediction and link prediction). Given the benefit of our graph-contextual training, we have a more principled way to obtain the protein representations from `LC-PLM-G` or ESM-2-G by averaging the embeddings of [BON] and [EON] (Note that to impose the inductive bias of the learned distribution of positive random walk paths to the embedding, we concat an [EDGE] token right after [EON] such that the output embeddings will be pulled towards the positive, i.e. existing graph context). In contrast, we can only obtain the protein embeddings from ESM-2 or `LC-PLM` by averaging the AA token embeddings, which are not very informative. We show the evaluation results on ogbn-proteins in Table 15 against a set of popular baselines. Our model achieves the best demonstration of the benefits of having graph context-aware protein embeddings learned in `LC-PLM-G`. We also provide ablation studies in Tables 17 and 18 where we choose two GNN backbones GCN and GraphSAGE to evaluate if the learned embeddings from `LC-PLM` and `LC-PLM-G` are beneficial to this task. The evidence shows that the learned embeddings consistently improve the performance and the graph context provides another boost.

As discussed in Appendix B, we can also perform more graph-specific self-supervised learning (Wang et al., 2021b;c;a; Zhao et al., 2022) within the given graph context before supervised fine-tuning. By using this, we may obtain better initialization for node embeddings which would potentially encode the graph context better and improve the final prediction performance.

| Model | Accuracy |
|---|---|
| Node2vec (Grover & Leskovec, 2016) | $0.6881 \pm 0.0065$ |
| GCN (Kipf & Welling, 2017) | $0.7251 \pm 0.0035$ |
| GraphSAGE (Hamilton et al., 2017) | $0.7443 \pm 0.0064$ |
| DeepGCN (Li et al., 2019) | $0.8496 \pm 0.0028$ |
| GAT (Veličković et al., 2017) | $0.8501 \pm 0.0046$ |
| DeeperGCN (Li et al., 2020) | $0.8580 \pm 0.0017$ |
| UniMP (Shi et al., 2020) | $0.8642 \pm 0.0008$ |
| GIPA (Li et al., 2023) | $0.8700 \pm 0.0010$ |
| ESM-2-G | $0.8920 \pm 0.0008$ |
| **LC-PLM-G** | $\mathbf{0.8925 \pm 0.0010}$ |

Table 15: Performance on `ogbn-proteins`.

| Model | Hits@100 |
|---|---|
| GraphSAGE (2017) | $0.1655 \pm 0.0240$ |
| GCN (2017) | $0.1867 \pm 0.0132$ |
| Node2vec (2016) | $0.2226 \pm 0.0083$ |
| NGNN (2021) + GCN | $0.3683 \pm 0.0099$ |
| NGNN (2021) + SAGE | $0.4005 \pm 0.0138$ |
| SEAL (2018) | $0.4880 \pm 0.0316$ |
| NGNN (2021) + SEAL | $0.5971 \pm 0.0245$ |
| ESM-2-G | $0.6092 \pm 0.0137$ |
| **LC-PLM-G** | $\mathbf{0.6150 \pm 0.0125}$ |

Table 16: Performance on `ogbl-ppa`.

| Model | Accuracy |
|---|---|
| GCN | $0.7251 \pm 0.0035$ |
| GCN+**LC-PLM** | $0.7643 \pm 0.0042$ |
| GCN+**LC-PLM-G** | $\mathbf{0.7668 \pm 0.0056}$ |
| GraphSAGE | $0.7443 \pm 0.0064$ |
| GraphSAGE + **LC-PLM** | $0.7662 \pm 0.0021$ |
| GraphSAGE + **LC-PLM-G** | $\mathbf{0.7679 \pm 0.0029}$ |

| Model | Hits@100 |
|---|---|
| GCN | $0.1867 \pm 0.0132$ |
| GCN+**LC-PLM** | $0.1946 \pm 0.0142$ |
| GCN+**LC-PLM-G** | $\mathbf{0.1988 \pm 0.0156}$ |
| GraphSAGE | $0.1655 \pm 0.0240$ |
| GraphSAGE + **LC-PLM** | $0.1847 \pm 0.0193$ |
| GraphSAGE + **LC-PLM-G** | $\mathbf{0.1876 \pm 0.0164}$ |

Table 17: Ablations on `ogbn-proteins`.    Table 18: Ablations on `ogbl-ppa`.

## APPENDIX I  PSEUDOCODE

We provide a detailed breakdown of our algorithm in this section and then we summarize this computation procedure into a pseudocode algorithmic block as shown in Algorithm 1. The algorithm procedure can be stated as follows:

**Input**  $\mathbf{T}_{l-1} : (B, S, D)$ tensor, where $B$ is batch size, $S$ is sequence length, and $D$ is hidden state dimension.

**Output**  $\mathbf{T}_l : (B, S, D)$ tensor.

**Process**  **1. Normalization**  $\mathbf{T}'_{l-1} : (B, S, D) \leftarrow \text{Norm}(\mathbf{T}_{l-1})$

    **2. Reversal**  $\hat{\mathbf{T}}'_{l-1} : (B, S, D) \leftarrow \text{Reverse}\left(\mathbf{T}'_{l-1}\right)$

    **3. Parallel Processing**  For both $\mathbf{T}'_{l-1}$ and $\hat{\mathbf{T}}'_{l-1}$, denoted as $\mathbf{T}'_{*,l-1}$:

      **a. Linear Transformations**

$$\mathbf{X}_{*,l-1} : (B, S, E) \leftarrow \text{Linear}^{\mathbf{X}_{*,l-1}}(\mathbf{T}'_{*,l-1})$$

$$\mathbf{Z}_{*,l-1} : (B, S, E) \leftarrow \text{Linear}^{\mathbf{Z}_{*,l-1}}(\mathbf{T}'_{*,l-1})$$

**b. Convolution and Activation** $\mathbf{X}'_{*,l-1} : (B, S, E) \leftarrow \mathrm{SiLU}(\mathrm{Conv1d}(\mathbf{X}_{*,l-1}))$

**c. Additional Linear Transformations**

$$\mathbf{B}_{*,l-1} : (B, S, N) \leftarrow \mathrm{Linear}^B_{*,l-1}(\mathbf{X}'_{*,l-1})$$

$$\mathbf{C}_{*,l-1} : (B, S, N) \leftarrow \mathrm{Linear}^C_{*,l-1}(\mathbf{X}'_{*,l-1})$$

**d. Delta Computation**

$$\Delta_{*,l-1} : (B, S, E) \leftarrow \log(1 + \exp(\mathrm{Linear}^\Delta_{*,l-1}(\mathbf{X}'_{*,l-1}) + \mathrm{Parameter}^\Delta_{*,l-1}))$$

**e. Parameter Scaling** $\bar{\mathbf{A}}_{*,l-1} : (B, S, E, N) \leftarrow \Delta_{*,l-1} \otimes \mathrm{Parameter}^{\bar{A}}_{*,l-1}$

**f. B Update** $\mathbf{B}_{*,l-1} : (B, S, E, N) \leftarrow \Delta_{*,l-1} \otimes \mathbf{B}_{*,l-1}$

**g. State Space Model Application**

$$\mathbf{Y}_{*,l-1} : (B, S, E) \leftarrow \mathrm{SSM}(\bar{\mathbf{A}}_{*,l-1}, \mathbf{B}_{*,l-1}, \mathbf{C}_{*,l-1})(\mathbf{X}'_{*,l-1})$$

**h. Final Computation** $\mathbf{Y}'_{*,l-1} : (B, S, E) \leftarrow \mathbf{Y}_{*,l-1} \odot \mathrm{SiLU}(\mathbf{Z}_{*,l-1})$

**4. Combination and Output** $\mathbf{T}_l : (B, S, D) \leftarrow \mathrm{Linear}^T(\mathbf{Y}'_{l-1} + \hat{\mathbf{Y}}'_{l-1}) + \mathbf{T}_{l-1}$

**Key Operations  Norm**  Normalization operation

**Linear**  Linear transformation

**Conv1d**  1D Convolution

**SiLU**  Sigmoid Linear Unit activation function

**SSM**  State Space Model

$\otimes$  Element-wise multiplication

$\odot$  Element-wise multiplication

---

**Algorithm 1 `BiMamba-S` Block**

---

**Input:** $\mathbf{T}_{l-1} : (B, S, D)$
**Output:** $\mathbf{T}_l : (B, S, D)$

1: $\mathbf{T}'_{l-1} : (B, S, D) \leftarrow \mathrm{Norm}(\mathbf{T}_{l-1})$
2: $\hat{\mathbf{T}}'_{l-1} : (B, S, D) \leftarrow \mathrm{Reverse}\left(\mathbf{T}'_{l-1}\right)$
3: **for** $\mathbf{T}'_{*,l-1} \in \{\mathbf{T}'_{l-1}, \hat{\mathbf{T}}'_{l-1}\}$ **do**
4:     $\mathbf{X}_{*,l-1} : (B, S, E) \leftarrow \mathrm{Linear}^{\mathbf{X}_{*,l-1}}(\mathbf{T}'_{*,l-1})$
5:     $\mathbf{Z}_{*,l-1} : (B, S, E) \leftarrow \mathrm{Linear}^{\mathbf{Z}_{*,l-1}}(\mathbf{T}'_{*,l-1})$
6:     $\mathbf{X}'_{*,l-1} : (B, S, E) \leftarrow \mathrm{SiLU}(\mathrm{Conv1d}(\mathbf{X}_{*,l-1}))$
7:     $\mathbf{B}_{*,l-1} : (B, S, N) \leftarrow \mathrm{Linear}^B_{*,l-1}(\mathbf{X}'_{*,l-1})$
8:     $\mathbf{C}_{*,l-1} : (B, S, N) \leftarrow \mathrm{Linear}^C_{*,l-1}(\mathbf{X}'_{*,l-1})$
9:     $\Delta_{*,l-1} : (B, S, E) \leftarrow \log(1 + \exp(\mathrm{Linear}^\Delta_{*,l-1}(\mathbf{X}'_{*,l-1}) + \mathrm{Parameter}^\Delta_{*,l-1}))$
10:     $\bar{\mathbf{A}}_{*,l-1} : (B, S, E, N) \leftarrow \Delta_{*,l-1} \otimes \mathrm{Parameter}^{\bar{A}}_{*,l-1}$
11:     $\mathbf{B}_{*,l-1} : (B, S, E, N) \leftarrow \Delta_{*,l-1} \otimes \mathbf{B}_{*,l-1}$
12:     $\mathbf{Y}_{*,l-1} : (B, S, E) \leftarrow \mathrm{SSM}(\bar{\mathbf{A}}_{*,l-1}, \mathbf{B}_{*,l-1}, \mathbf{C}_{*,l-1})(\mathbf{X}'_{*,l-1})$
13:     $\mathbf{Y}'_{*,l-1} : (B, S, E) \leftarrow \mathbf{Y}_{*,l-1} \odot \mathrm{SiLU}(\mathbf{Z}_{*,l-1})$
14: **end for**
15: $\mathbf{T}_l : (B, S, D) \leftarrow \mathrm{Linear}^T(\mathbf{Y}'_{l-1} + \hat{\mathbf{Y}}'_{l-1}) + \mathbf{T}_{l-1}$
16: **return** $\mathbf{T}_l$

---

## APPENDIX J  SAMPLING BIAS IN RANDOM WALK

For random walk sampling, there are two parameters $p$ and $q$ we can use to control the direction of exploration (a balance between the depth-first search (DFS) and breath-first search (BFS)).

**Return Parameter** $p$  Parameter $p$ controls the likelihood of immediately revisiting a node. A high value ($p > \max(q, 1)$) reduces the chance of sampling an already-visited node, encouraging moderate exploration and avoiding 2-hop redundancy.

**In-out Parameter** $q$   Parameter $q$ allows the search to differentiate between "inward" and "outward" nodes:

- If $q > 1$, the walk is biased towards nodes close to node $t$, approximating BFS behavior.

- If $q < 1$, the walk tends to visit nodes further from node $t$, encouraging outward exploration similar to DFS.

**Benefits over Pure BFS/DFS**   Random walks offer several advantages over traditional BFS/DFS approaches:

1. **Computational Efficiency:** They are efficient in terms of both space and time requirements.

2. **Scalability:** The space complexity to explore the immediate neighbors of every node is $O(|E|)$ for a graph with edge set $E$.

3. **Flexibility:** For 2nd order random walks, the space complexity becomes $O(a^2|V|)$, where $a$ is the average degree of the graph and $V$ is the vertex set.

4. **Effective Sampling:** Random walks provide a convenient mechanism to increase the effective sampling rate by reusing samples across different source nodes.

5. **Parallel Sampling:** Due to the Markovian nature of the random walk, $k$ samples for $l - k$ nodes can be generated at once, resulting in an effective complexity of $O(\frac{l}{k(l-k)})$ per sample.

This approach combines the benefits of BFS and DFS, allowing for a more nuanced exploration of network structures that exhibit both structural equivalence and homophily.

## APPENDIX K   TRAINING AND EVALUATION CURVES

### K.1   THE FIRST-STAGE PRETRAINING

We show the training and evaluation loss curves in Figure 9 for our first-stage pretraining of **LC-PLM** on 100B `UniRef50`. This pretrained model is used in all downstream task evaluations reported in the paper.

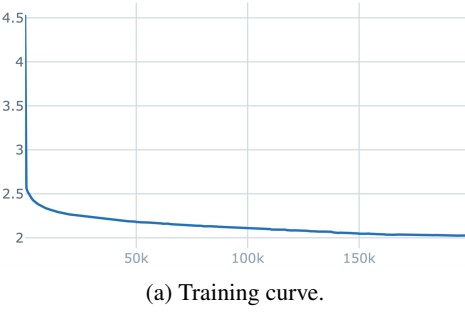
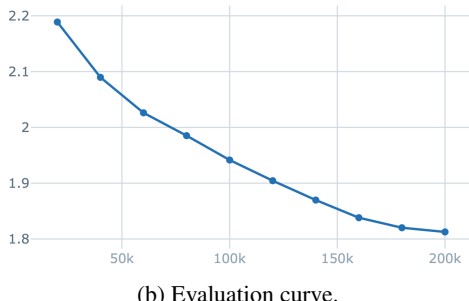

(a) Training curve.   (b) Evaluation curve.

Figure 9: The first-stage pretraining of 790M **LC-PLM** on 100B `UniRef50`. The evaluation set is 250K `UniRef90`.

### K.2   THE SCALING LAW EXPERIMENTS

We show the training and evaluation loss curves in Figure 10, Figure 11, Figure 12, and Figure 13 for our scaling law training of 130M, 370M, 790M, and 1.4B **LC-PLM** on 20B `UniRef90`. The evaluation set is the held-out 250K `UniRef90`.

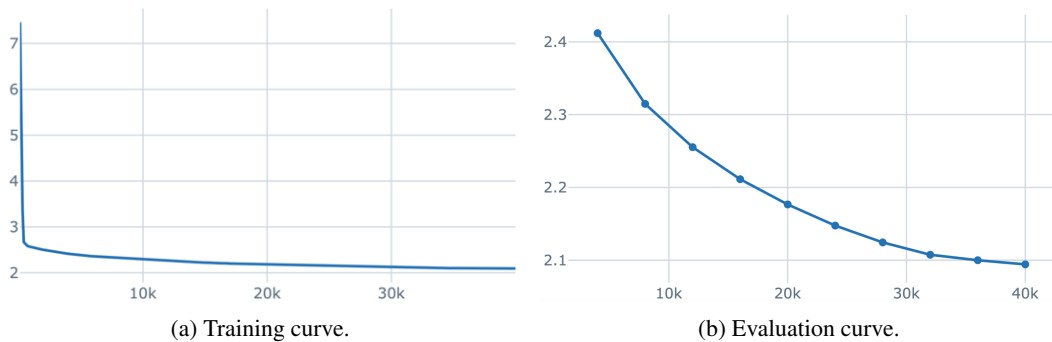

(a) Training curve.          (b) Evaluation curve.

Figure 10: The scaling law training of 130M `LC-PLM` on 20B `UniRef90`. The evaluation set is 250K `UniRef90`.

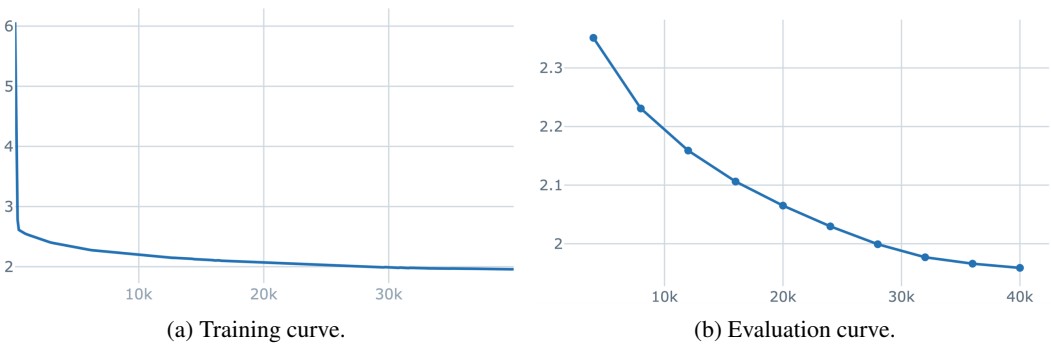

(a) Training curve.          (b) Evaluation curve.

Figure 11: The scaling law training of 370M `LC-PLM` on 20B `UniRef90`. The evaluation set is 250K `UniRef90`.

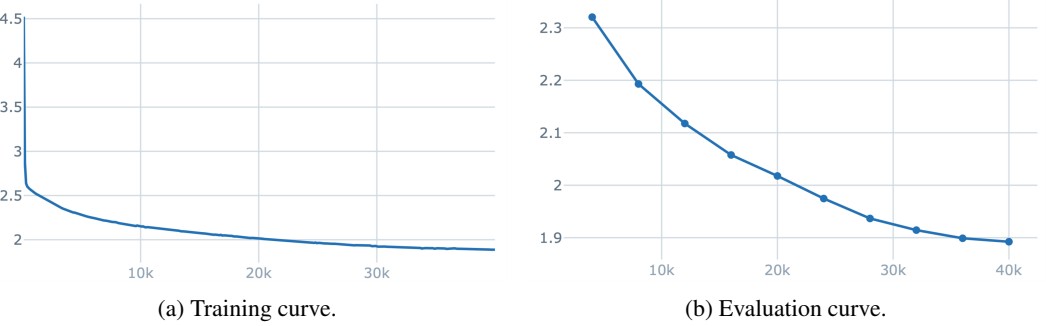

(a) Training curve.          (b) Evaluation curve.

Figure 12: The scaling law training of 790M `LC-PLM` on 20B `UniRef90`. The evaluation set is 250K `UniRef90`.

### K.3 THE LENGTH EXTRAPOLATION EXPERIMENTS

We show the training and evaluation loss curves in Figure 14, Figure 15, and Figure 16 for our scaling law training of 130M, 370M, and 790M `LC-PLM` on the 128-256 bin of `UniRef90`. The evaluation set is the held-out 250K `UniRef90`.

### K.4 THE SECOND-STAGE GRAPH CONTEXTUAL TRAINING

We show the training loss curves in Figure 17a, and Figure 17b for our second-stage graph contextual training of 790M `LC-PLM-G` on protein sequences included in `ogbn-proteins` and `ogbl-ppa`. The evaluation set is the held-out 250K `UniRef90`.

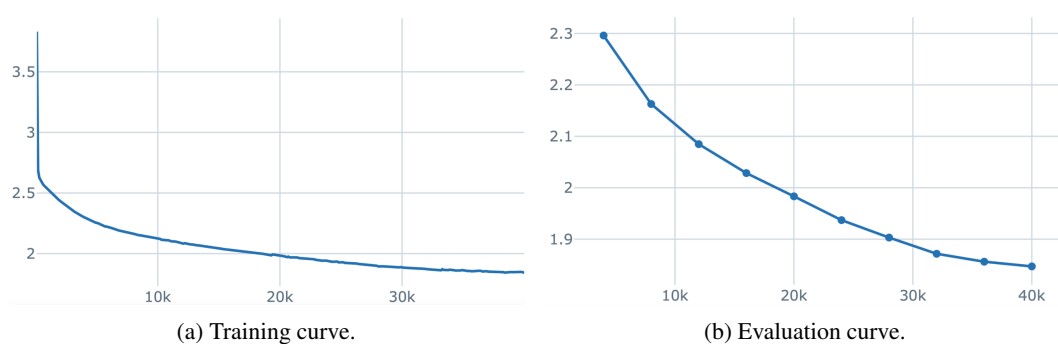

(a) Training curve.
(b) Evaluation curve.

Figure 13: The scaling law training of 1.4B `LC-PLM` on 20B `UniRef90`. The evaluation set is 250K `UniRef90`.

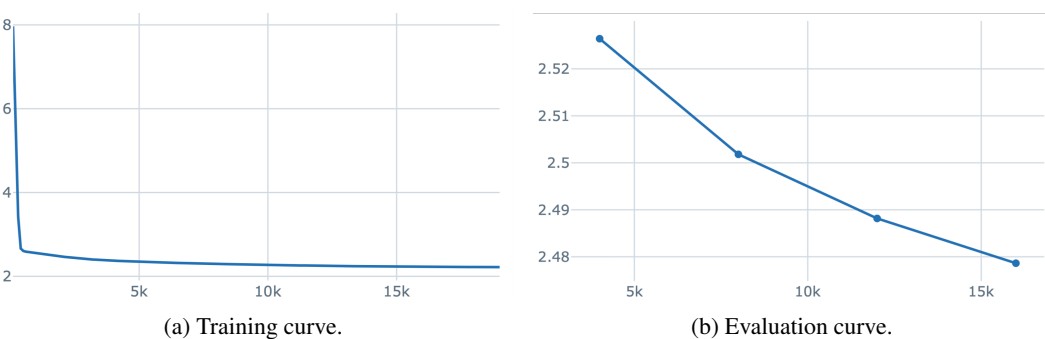

(a) Training curve.
(b) Evaluation curve.

Figure 14: The length extrapolation training of 130M `LC-PLM` on the 128-256 bin of `UniRef90`. The evaluation set is 250K `UniRef90`.

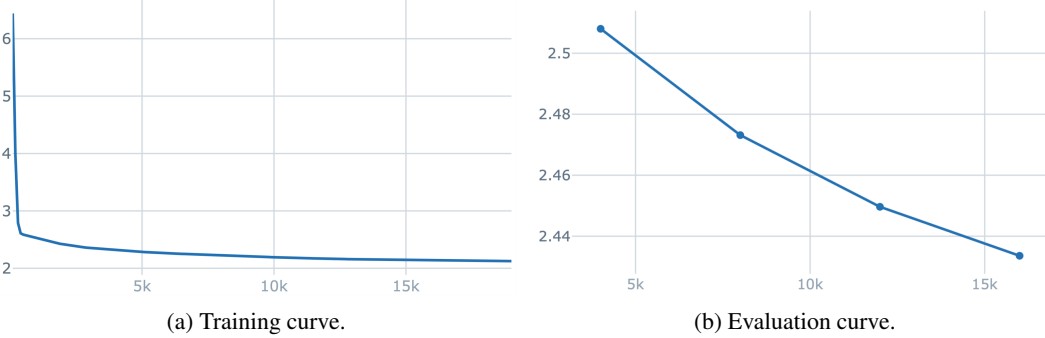

(a) Training curve.
(b) Evaluation curve.

Figure 15: The length extrapolation training of 370M `LC-PLM` on the 128-256 bin of `UniRef90`. The evaluation set is 250K `UniRef90`.

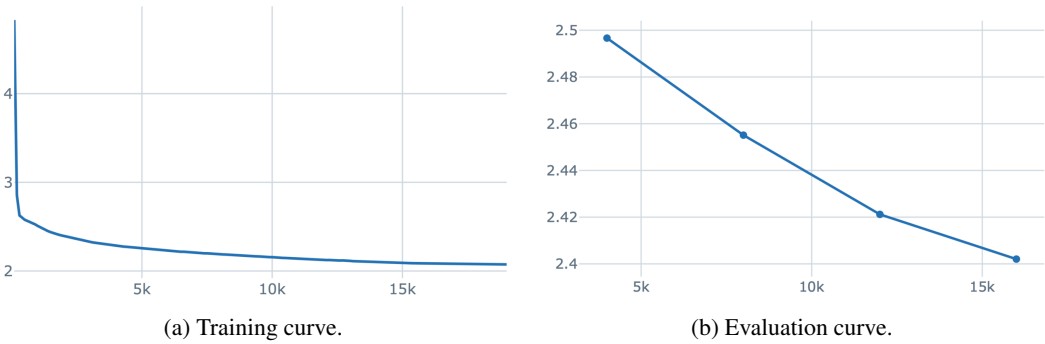

(a) Training curve.
(b) Evaluation curve.

Figure 16: The length extrapolation training of 790M `LC-PLM` on the 128-256 bin of `UniRef90`. The evaluation set is 250K `UniRef90`.

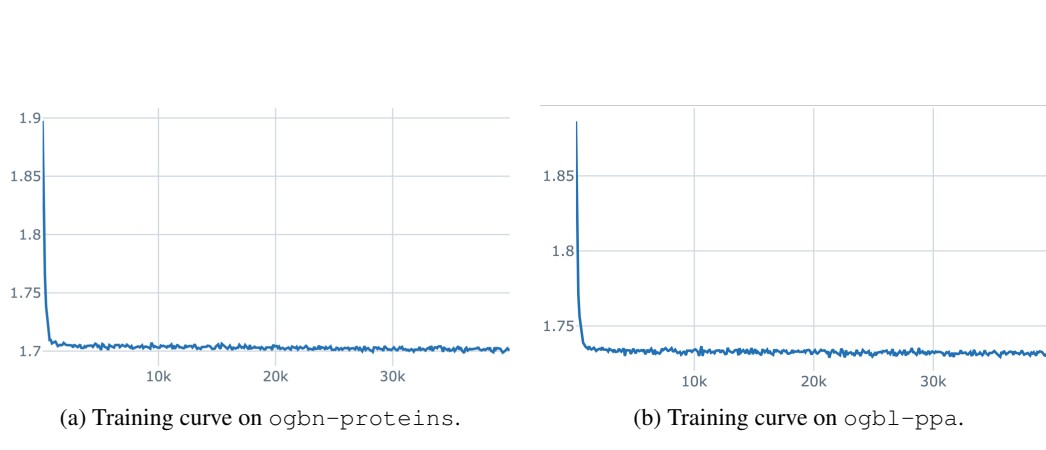

(a) Training curve on `ogbn-proteins`.

(b) Training curve on `ogbl-ppa`.

Figure 17: The second-stage training of 790M **LC-PLM-G** on `ogbn-proteins` and `ogbl-ppa`. The evaluation set is 250K `UniRef90`.

