# OpenReview forum: "Long-context Protein Language Model"
_ICLR.cc/2025/Conference — Submitted to ICLR 2025_

### Official Review · Reviewer_jUT5 · 2024-11-01

**Soundness:** 2
**Presentation:** 3
**Contribution:** 1
**Rating:** 3
**Confidence:** 4

**Summary:**

The authors propose a method for masked-modeling of protein sequences with large context sizes.

**Strengths:**

- Generally important topic of modeling proteins
- Diverse, extensive and involved experiments done
- Good presentation of the material

**Weaknesses:**

- Originality: This method already exists as ProtMamba [1]. ProtMamba also uses Mamba for protein modeling and introduces long context tasks. The authors should propose a new method to model protein sequences and long-contexts.

- Significance:
a) The significance of this work is also diminished by many false and tenuous claims. E.g. "Such protein LMs cannot extrapolate to longer proteins [..]": this is not true, and cannot and has not been shown theoretically nor empirically. Transformer-based pLMs can extrapolate to longer contexts and can also handle relatively long contexts (e.g. 16k context size). Also "they fail to account for the underlying biological mechanisms [...]" is just a tenuous claim without any evidence. The claim that"our work [...] learns universal AA token level presentation" also bears any evidence and any definition what "universal" means. The authors should remove all tenuous and false claim (not limited to the ones I mentioned) from the manuscript.

b) The approach with graph-contextual modeling is ad-hoc. It is unclear why proteins from a PPI graph should be relevant contexts for a particular protein at hand. The homology-criterion that is used by [1] is much better justified and the default approach to assemble long-contexts.

- Technical errors:
Comparisons with ProtMamba and Transformer based architectures are missing. It seems that the authors have only trained their own architecture in this work and not a single model based on another architecture. The authors should at least compare with ProtMamba, and with 1-2 Transformer-based architectures. Therefore, it also remains unclear from where the alleged performance gains arise: the increased context, the graph-style pre-trained, the architecture, or any other component.


Other flaws:
Table 1 and table 3: one digit too much is displayed.


References:
[1] Sgarbossa, D., Malbranke, C., & Bitbol, A. F. (2024). ProtMamba: a homology-aware but alignment-free protein state space model. bioRxiv, 2024-05.

**Questions:**

- Can you clearly state which models have been trained in this work and for which models you just took pretrained versions?

---

> ### Author Response · Authors · 2024-11-22
> **Response to Reviewer jUT5 [1/n]**
>
> We thank the reviewer for their feedback on our paper. We performed multiple novel ablations and comparisons that we report in the main comment of the rebuttal. We also address the reviewer’s specific comments and concerns below:
>
> >**1. Originality and comparison to ProtMamba.**
>
> We respectfully disagree with the assertion of “This method already exists as ProtMamba”. We would like to clarify several key distinctions:
>
> - We have **discussed the key differences between our work and ProtMamba [1] in lines 152-153 and 158-161 and Table 1**. To summarize again, *ProtMamba trained on concatenated homologous protein sequences* with autoregressive causal language modeling and an infilling objective *focusing on protein sequence generation*. However, our LC-PLM focuses on *learning a foundation-level long context protein language model (pLM)* that can provide universal amino acid level protein representations for extremely long protein sequences; protein complexes, multimers, and heterodimers with encoded protein interaction context information. We would like to **demonstrate that LC-PLM is a much better foundation pLM than the previous open-sourced SOTA foundation pLMs (i.e. ESM-2 [[Lin et al.](https://www.biorxiv.org/content/10.1101/2022.07.20.500902v1.full.pdf)] and CARP [[Yang et al.](https://www.biorxiv.org/content/10.1101/2022.05.19.492714v2.full.pdf)])**. We additionally argue that, unlike ProtMamba [[Sgarbossa et al.](https://www.biorxiv.org/content/10.1101/2024.05.24.595730v1)], which uses homologous sequences as context, our LC-PLM-G proposed a novel way to encode protein interaction graph information that contextualizes functionally related proteins rather than semantically similar ones.
>
> - Here we provide new experimental results directly comparing our LC-PLM to ProtMamba. In our new experiments, we evaluate ProtMamba on the downstream tasks we used in our paper. **The performance of ProtMamba is much lower than LC-PLM (even CARP and ESM-2 as shown in general response) across all tasks**. This suggests that ProtMamba pretrained with concatenated homologous protein sequences potentially leads to degraded representations of individual protein sequences. After all, ProtMamba is trained to use homologous sequences as context for protein generation tasks (e.g. infilling and fitness prediction), rather than producing useful representations for protein sequences. Also it is worth noting that **ProtMamba cannot extrapolate to sequence > 2048** since they use positional encodings with fixed length in training. We summarized the results in the tables below. We added these results to our manuscript. Also as a side note, [ProtMamba](https://openreview.net/forum?id=BMfHO2lXGe) is a concurrent submission to ICLR 2025 so other submissions can be excused for not comparing against them according to [ICLR’s policy](https://iclr.cc/Conferences/2025/FAQ).

---

> ### Author Response · Authors · 2024-11-22
> **Response to Reviewer jUT5 [2/n]**
>
> Table A. Protein structure prediction with LMFold. Structure prediction performance (TM score) are reported on different hold out datasets.
> | Models               | CASP15-multimers                               | CASP14                               | Benchmark2                         |
> |-----------------------|-----------------------------------------------|--------------------------------------|------------------------------------|
> | LC-PLM-790M (100B)   | **0.5109 ± 0.0070**                           | **0.4154 ± 0.0080**                 | **0.6290 ± 0.0071**               |
> | ProtMamba-public     | N/A (ProtMamba cannot run on protein sequences with > 2048 length) | 0.3288 ± 0.0091                 | 0.4515 ± 0.0062               |
>
> Table B.  Evaluation on TAPE tasks in supervised fine-tuning setting. We report the top-1 accuracy for the Remote Homology fold-level test set; accuracy for the 3-class secondary structure prediction on the CB513 test set; Spearman’s correlation coefficients for the test sets for the Stability and Fluorescence prediction tasks. For the Jacobian contact map prediction task, we adopted the methods from [[Zhang et al.](https://www.biorxiv.org/content/10.1101/2024.01.30.577970v1)] to use categorical Jacobian matrices computed from protein language models as the zero-shot prediction for protein contact maps and report the precision@2/L (L is the length of a protein sequence) on the validation set of ProteinNet dataset [[AlQuraishi](https://bmcbioinformatics.biomedcentral.com/articles/10.1186/s12859-019-2932-0)].
> | Models               | PPI Graph       | Jacobian Contact Map | Remote Homology     | Secondary Structure | Stability       | Fluorescence    |
> |-----------------------|-----------------|-----------------------|---------------------|---------------------|-----------------|-----------------|
> | LC-PLM-790M (100B)   | None            | 47.1                 | 35.14 ± 1.69        | **85.07 ± 0.03**    | 0.794 ± 0.003 | 0.692 ± 0.002   |
> | LC-PLM-G-790M (100B) | ogbn-proteins   | 47.15                | **35.74 ± 0.93**        | 85.02 ± 0.11    | **0.801 ± 0.001** | **0.709 ± 0.033**   |
> | LC-PLM-G-790M (100B) | ogbl-ppa        | **47.23**            | 35.60 ± 1.45        | 85.01 ± 0.03    | **0.801 ± 0.001** | 0.693 ± 0.002   |
> | ProtMamba-public     | None            | 10.96                | 17.82 ± 1.85        | 68.43 ± 0.06        | 0.726 ± 0.012    | 0.688 ± 0.005   |

---

> ### Author Response · Authors · 2024-11-22
> **Response to Reviewer jUT5 [3/n]**
>
> - Also, we want to note that unlike ProtMamba, we propose to use BiMamba-S, which is the most proper way to realize bi-directionality in Mamba. **BiMamba-S can be formulated in a more theoretical way as structured SSMs with quasi-separable matrices [[Hwang et al.](https://arxiv.org/abs/2407.09941)]**. We did not apply quasi-separable mixers in our model since **Mamba currently has a much better software-hardware interface environment and distributed training support in practical implementations that help us train a large foundation-level model feasibly and efficiently**. We added this discussion in our paper as well. We also want to argue that, just like Transformer, which has been used in numerous impactful works such as GPT and ESM-2, BiMamba-S also serves as a versatile and effective architectural choice for sequence modeling. The reuse of proven architectures like Transformer or BiMamba does not diminish the novelty of a work. Instead, it allows researchers to focus on solving domain-specific challenges (in our case, exploring the capability of building up a new type of foundation pLM based on BiMamba-S). Our work follows this principle, leveraging this architecture to demonstrate its effectiveness in pushing the boundaries of foundation pLM.
>
> - Besides the architectural choice based on Mamba, we have **many other important contributions to this field as summarized in the Introduction**: (1) method to encode biological interaction information into pLM, we propose a novel second-stage training based on random walks over graphs to extend the long-context capabilities of LC-PLM to leverage the PPI graph context; (2) we demonstrate that LC-PLM has improved length extrapolation capabilities, favorable scaling laws, and achieved a 7% to 34% improvement on downstream tasks (e.g. protein structure prediction (CASP15-multimers, CASP14, Benchmark2), tasks in TAPE and ProteinGym) compared to ESM-2, CARP, especially for longer proteins and protein complexes; (3) we demonstrate its effectiveness in capturing graph contextual information on remote homology detection (TAPE) in Table 3, protein function prediction (ogbn-proteins) in Table 14 and 16, and PPI link prediction (ogbl-ppa) in Table 15 and 17.
>
> >**2. Significance.**
>
> We respectfully disagree with the assertion of “The significance of this work is also diminished by many false and tenuous claims”.
>
> - *"Such protein LMs cannot extrapolate to longer proteins [..]": this is not true, and cannot and has not been shown theoretically nor empirically. Transformer-based pLMs can extrapolate to longer contexts and can also handle relatively long contexts (e.g. 16k context size).*
>
> Since transformers are invariant to input position, they usually use positional encodings along with the sequence input. These encodings can be difficult to extend past the maximum length seen during training. Most transformer-based foundation pLMs limit the input length during pretraining such that they cannot extrapolate during inference. For example, ESM-2 has a maximum input length of 1024 residues; and it performs worse on both shorter and longer sequences [https://github.com/facebookresearch/esm/discussions/76]. We also provide evidence in Table 6 that ESM-2 fails to extrapolate on both longer (2k-8k) and shorter (0-128) sequences. It is worth noting that ESM-2 even used extrapolatable positional encodings [[Zhao et al.](https://www.google.com/url?sa=t&source=web&rct=j&opi=89978449&url=https://arxiv.org/abs/2312.17044&ved=2ahUKEwjfuqnNzfCJAxWAMlkFHeDgABUQFnoECBYQAQ&usg=AOvVaw0fldeIoSBJQd5fNgnKjOAN)], i.e. RoPE [[Su et al.](https://www.google.com/url?sa=t&source=web&rct=j&opi=89978449&url=https://arxiv.org/abs/2104.09864&ved=2ahUKEwi08O3jzfCJAxWgGlkFHQIeNgkQFnoECAwQAQ&usg=AOvVaw2CO33TcxkzYh5uPFyjwCsf)], which however performs poorly for protein sequences. CARP as a new type of attention-free foundation pLM increased the input sequence length to 4k but performs worse than ESM-2 due to its lack of expressiveness. Our LC-PLM built off BiMamba-S can take in infinite-length input sequences and extrapolate well on both shorter and longer sequences; and meanwhile provide even better performance on regular length proteins across various downstream tasks.

---

> ### Author Response · Authors · 2024-11-22
> **Response to Reviewer jUT5 [4/n]**
>
> - *Also "they fail to account for the underlying biological mechanisms [...]" is just a tenuous claim without any evidence.*
>
> We provide much evidence throughout the paper in Tables 2, 3, 4 to show transformer-based SOTA pLM (i.e. ESM-2) fails to perform well when the task requires knowledge and information about biomolecular interactions and dynamics. For example, in Table 2, we show that ESM-2 performs much worse than LC-PLM on all folding benchmarks, especially on protein complexes, which need protein interaction information to better infer the structures; in Table 3, we show that ESM-2 also performs much worse on TAPE tasks, i.e. remote homology prediction and secondary structure prediction; in Table 4, we show that ESM-2 is also much worse on ProteinGym benchmark. We think there is sufficient evidence to demonstrate the suboptimality of transformer-based SOTA pLM (ESM-2) in capturing biomolecular interactions and dynamics and accounting for such underlying biological mechanisms.
>
> - *The claim that"our work [...] learns universal AA token level presentation" also bears any evidence and any definition what "universal" means.*
>
> We use the term “universal” since (1) this term has been widely-used in protein representation learning literature [[Alley et al.](https://pubmed.ncbi.nlm.nih.gov/31636460/), [Detlefsen et al.](https://www.nature.com/articles/s41467-022-29443-w)], where researchers define pLM is *“[learning universal, cross-family representations of protein space](https://www.nature.com/articles/s41467-022-29443-w)”*; and (2) we pretrain our models on the **Universal** Protein Reference Clusters (UniRef) dataset which contains universal protein sequence resources. Given such learned high-quality protein embeddings, we can achieve decent performance across a variety of downstream tasks, which demonstrates the universality of such protein representations.
>
> >**3. Why don’t we use homology-criterions like in ProtMamba?**
>
> Thank you for this question. We don’t consider homology-criterion because the context contained in homology representations, such as multiple sequence alignment (MSA) used for pretraining ProtMamba, are already contained in the sequences of individual proteins. After all, MSAs can be considered as performing clustering on a set of protein sequences. As such, we think homology does not bring additional information beyond protein sequences. On the other hand, we consider biological graphs such as PPI graphs, which contain functional interactions between protein sequences derived from additional sources such as annotations and wet lab biological experiments.
>
> >**4. Train foundation pLMs with other Transformer architectures for comparison.**
>
> In our scaling law, length extrapolation, structure prediction, TAPE benchmark, and ProteinGym experiments, we pretrained our own Transformers/ESM-2 on the same dataset with the same number of tokens across different sizes from scratch as baseline comparisons. We clearly indicated where we retrained models / used publicly pretrained models in our manuscripts, e.g. lines 338-341, lines 366-368, lines 433-434, the footnote in Table 2, etc.
>
> We want to note that our main goal is **building up a foundation-level long context protein language model (pLM)** that can provide universal amino acid level protein representations for extremely long protein sequences; protein complexes, multimers, and heterodimers with encoded protein interaction context information, not debating on selecting/tailoring a specific architecture, just like ESM-2 (there exist many other decent linear RNNs/Transformers architectures in the community but trying out all of them is beyond the scope of our paper). We would like to **demonstrate that LC-PLM is a much better foundation pLM than the previous open-sourced SOTA foundation pLMs (i.e. ESM-2 and CARP)**. We compare our LC-PLM to them in Table 3. We provide more comparisons in the Table below.

---

> ### Author Response · Authors · 2024-11-22
> **Response to Reviewer jUT5 [5/n]**
>
> | Models               | PPI Graph       | Jacobian Contact Map | Remote Homology     | Secondary Structure | Stability (spearman rho) | Fluorescence (spearman rho) |
> |-----------------------|-----------------|-----------------------|---------------------|---------------------|---------------------------|-----------------------------|
> | ESM-2-650M (100B)    | None            | 44.05                | 26.57 ± 0.49        | 79.86 ± 0.09        | 0.763 ± 0.008             | 0.695 ± 0.002              |
> | ESM-2-G-650M (100B)  | ogbn-proteins   | 32.35                | 25.60 ± 0.77        | 79.76 ± 0.24        | 0.750 ± 0.016             | 0.694 ± 0.002              |
> | ESM-2-G-650M (100B)  | ogbl-ppa        | 26.66                | 27.18 ± 0.63        | 79.91 ± 0.24        | 0.753 ± 0.009             | 0.693 ± 0.001              |
> | LC-PLM-790M (100B)   | None            | 47.1                 | 35.14 ± 1.69        | **85.07 ± 0.03**    | 0.794 ± 0.003             | 0.692 ± 0.002              |
> | LC-PLM-G-790M (100B) | ogbn-proteins   | 47.15                | **35.74 ± 0.93**    | 85.02 ± 0.11        | **0.801 ± 0.001**         | **0.709 ± 0.003**          |
> | LC-PLM-G-790M (100B) | ogbl-ppa        | **47.23**            | 35.60 ± 1.45        | 85.01 ± 0.03        | **0.801 ± 0.001**         | 0.693 ± 0.002              |
> | ProtMamba-public            | None            | 10.96                | 17.82 ± 1.85        | 68.43 ± 0.06        | 0.726 ± 0.012             | 0.688 ± 0.005              |
> | CARP-640M-public     | None            | 25.83                | 28.0 ± 0.8          | 83.0 ± 0.1          | 0.72 ± 0.01               | 0.68 ± 0.002               |
> | ESM2-650M-public (1T, for reference only)    | None            | 66.85            | 33.43 ± 0.35    | 84.30 ± 0.15        | 0.804 ± 0.006         | 0.688 ± 0.001              |
>
> >**5. Can you clearly state which models have been trained in this work and for which models you just took pretrained versions?**
>
> We clearly indicated where we retrained models / used publicly pretrained models in our manuscripts, e.g. lines 338-341, lines 366-368, lines 433-434, the footnote in Table 2, etc. In summary, all ESM-2(-G) and LC-PLM(-G) models with 100B tokens are retrained; all models with “-public” postfix are taken from public pretrained checkpoints. We also provide training and dataset details in Appendices D and E.

---

> ### Author Response · Authors · 2024-11-25
> **Kindly reminder of the end of rebuttal**
>
> Dear Reviewer jUT5,
>
> As the discussion period is coming to an end tomorrow, we kindly ask you to review our response to your comments and let us know if you have any further queries. Alternatively, if you think we addressed your concerns properly and could raise the rating of the paper, we would be extremely grateful. We eagerly anticipate your response and are committed to addressing any remaining concerns before the discussion period concludes.
>
> Best regards,
>
> Authors

---

> ### Comment · Reviewer_jUT5 · 2024-11-25
> **Discussion**
>
> The differences to ProtMamba are just in the collection of training data and -- as the authors state -- in the focus on a foundation-level long context protein model. Also ProtMamba can be seen as a foundation level-protein model. The approach is still almost identical to ProtMamba (pre-print available since May). The adaption of a general architecture (such as Transformer, Mamba, etc) to a new application domain would be a relevant application paper, but this step has already been done. The rebuttal has again tenuous claims such as "ProtMamba cannot extrapolate to sequence > 2048", where it is clearly shown to perform well even for context sizes of 2^17. The claimed differences are overall too minor to represent an new ML application.
>
> I appreciate the authors enthusiasm about their method and the effort that went into this rebuttal, but I decide to keep my score.

---

> > ### Author Response · Authors · 2024-11-25
> > **Reply to Reviewer jUT5**
> >
> > Thank you for your feedback. We hope you were able to review our response and acknowledge that we did perform additional experiments as requested.
> >
> > > The differences to protmamba are just in the collection of training; the method is identical to ProtMamba; the adaption has been done in ProtMamba.
> >
> > We have tons of differences/contributions more than ProtMamba that we discussed in the rebuttal (https://openreview.net/forum?id=Et0SIGDpP5&noteId=q7eSAEetLN) and our paper. We summarized again as follows:
> > - We introduced a novel method to encode biological interaction information into pLM, i.e. a novel second-stage training based on random walks over graphs to extend the long-context capabilities of LC-PLM to leverage the PPI graph context; we discussed this in the rebuttal (https://openreview.net/forum?id=Et0SIGDpP5&noteId=q7eSAEetLN) and our paper;
> > - In terms of architectural design, we carefully studied (1) bidirectionally of Mamba blocks, (2) shared projection layers in bidirectional Mamba blocks; (3) untied input/output embedding layers to improve the uniformity of embeddings, which are what ProtMamba didn’t study; we discussed this in our paper;
> > - We demonstrate that LC-PLM has improved length extrapolation capabilities, favorable scaling laws, and achieved a 7% to 34% improvement on downstream tasks (e.g. protein structure prediction (CASP15-multimers, CASP14, Benchmark2), tasks in TAPE and ProteinGym) compared to ESM-2, CARP, especially for longer proteins and protein complexes; we discussed this in the rebuttal (https://openreview.net/forum?id=Et0SIGDpP5&noteId=q7eSAEetLN) and our paper;
> > - We demonstrate its effectiveness in capturing graph contextual information on remote homology detection (TAPE) in Table 3, protein function prediction (ogbn-proteins) in Table 14 and 16, and PPI link prediction (ogbl-ppa) in Table 15 and 17; we discussed this in the rebuttal (https://openreview.net/forum?id=Et0SIGDpP5&noteId=q7eSAEetLN) and our paper;
> > - In terms of pretraining data used for the 2nd stage, we believe graph of sequences is a more generalizable data structure as it subsume 1) set of sequences and 2) sequence of sequences; we discussed this in the rebuttal (https://openreview.net/forum?id=Et0SIGDpP5&noteId=9ZP4M9VrFd) and our paper;
> > - Also as a side note, ProtMamba (https://openreview.net/forum?id=BMfHO2lXGe) is a concurrent submission to ICLR 2025 and just an online preprint so other submission’s contributions should not be undermined according to ICLR’s policy (https://iclr.cc/Conferences/2025/FAQ); we mentioned this in the rebuttal (https://openreview.net/forum?id=Et0SIGDpP5&noteId=q7eSAEetLN); Despite this, we added direct comparison to them in the rebuttal and our method is beating ProtMamba across all tasks (e.g. outperforming ProtMamba by up to 331% on contact map), as shown in the Tables A and B of the rebuttal (https://openreview.net/forum?id=Et0SIGDpP5&noteId=sMTKYS7SUe; and https://openreview.net/forum?id=Et0SIGDpP5&noteId=JC3OWeSLe7), and Tables 2 and 3 in our revised manuscript.
> >
> > > ProtMamba is a foundation model.
> >
> > We disagreed. Since ProtMamba is pretrained with concatenated homolougous sequences from MSA, it is expected to perform well on tasks where the inputs are a set of homologous sequences, such as deep mutational scan data. However, as our experiments show (in Tables A and B from our rebuttal, in Tables 2 and 3 from our revised manuscript), it suffers on other tasks where the input format is individual proteins, including four TAPE tasks and protein structure prediction tasks. A foundation level-protein model should perform well on all these protein property prediction tasks and can be used as a building block for specialized tasks such as mutational fitness prediction (e.g. [TransceptEVE](https://www.biorxiv.org/content/10.1101/2022.12.07.519495v1.full.pdf)) and structure prediction (e.g. [ESMFold](https://www.science.org/doi/10.1126/science.ade2574)).
> >
> > > Tenuous claim “ProtMamba cannot extrapolate to sequence > 2048”.
> >
> > As we pointed out in our rebuttal, ProtMamba used positional encodings (PEs) (https://github.com/Bitbol-Lab/ProtMamba-ssm/blob/8befff756b2db7b6dc56d0a07163eb02e27b2731/ProtMamba_ssm/modules.py) which makes it unable to extrapolate to sequences > 2048 on any downstream tasks that need fine-tuning on longer sequences. Also ProtMamba used learnable PE (https://github.com/Bitbol-Lab/ProtMamba-ssm/blob/8befff756b2db7b6dc56d0a07163eb02e27b2731/ProtMamba_ssm/modules.py#L464) as an improper design choice that will do harm to the length extrapolation capability of the model since such PE has been demonstrated as an un-extrapolatable PE in many literature [[Zhao et al.](https://arxiv.org/abs/2312.17044); [Sun et al.](https://aclanthology.org/2023.acl-long.816.pdf)].
> >
> > We respectfully ask the reviewer to **stop using terms like “tenuous claims” unless they carefully read our paper and rebuttal**. Have a great day.

---

### Official Review · Reviewer_8QMc · 2024-11-03

**Soundness:** 2
**Presentation:** 2
**Contribution:** 2
**Rating:** 3
**Confidence:** 4

**Summary:**

The paper presents LC-PLM, a protein language model based on a bidirectional selective structured state-space models with shared projection layers (BiMamba-S) for learning protein representations. Additionally, it introduces a graph-based variant, LC-PLM-G, incorporating protein-protein interaction (PPI) graphs for training and show learning paradigm with extended context beyond individual proteins. The authors claim that (1) LC-PLM demonstrates better length extrapolation capabilities and outperforms Transformer-based models like ESM-2 on various essential protein-related tasks; (2) LC-PLM-G shows promise in protein structure and function prediction tasks by integrating biological interaction contexts.

**Strengths:**

- The paper presents a range of experimental evaluations across downstream tasks, showing better performance compared to one standard Transformer-based protein language model ESM-2. The evaluation tasks are diverse and comprehensive.
- The introduction of LC-PLM-G, incorporating PPI graphs, is a novel direction for protein language models. Using relational information between proteins has potential for realistic applications where protein interactions play a crucial role.

**Weaknesses:**

- The paper does not adequately justify the selective structured state-space (S4) models over existing Transformers-based PLMs (eg. ESM-2), which are widely adopted for protein sequence modeling. Protein sequence understanding does not usually face the same latency/bandwidth constraints as real-time LLM deployments such as ChatGPT, where SSMs could possibly bring more practical value with linear inference time. Without clear advantages or innovations in the application of SSMs, this choice lacks strong motivation to stand convincing.
- The paper assumes that longer context modeling is inherently beneficial for protein sequences. However, the biological need for extremely long contexts remains unclear in many protein-related applications, especially since functional motifs often reside within local regions of a protein rather than requiring entire sequence context. Without a clear biological or empirical rationale, the assumption that long contexts are “essential” for protein understanding is questionable to me.
- The authors highlight LC-PLM’s computational efficiency over Transformers due to the quadratic complexity of Transformers. However, given that training/inference-efficient Transformers exist (flash attention for example), the emphasis on efficiency without practical demand diminishes the paper’s significance.
- Across the evaluation tasks (yes i agree they are many), only ESM-2 is there for comparison while for many of the task ESM-2 (alone) is not the real state-of-the-art (SOTA). Note that there are many interesting works built based on the ESM-2 embeddings and give very promising SOTA results for function prediction or mutation effect modeling. I wonder how the LC-PLM embedding accommodate these existing model and perform better than ESM-2.

**Questions:**

- In RQ1, how do the authors hold out the test 250k sequence? Do the authors re-train the ESM2 for plotting the Figure 4? Moreover, in Figure 4/5, what is in specific the “Evaluation Loss” for the label of the y-axis (vertical direction)?
- When reporting folding evaluation, how do the structures obtained from ESM-2?  My concern is, for evaluation fairness, one should align the folding trunk while on the other hand, the distribution over the residue-level embeddings from (1) LC-PLM and (2) ESM-2 can be different  in the vector space. Please elaborate or point out some relevant context to address this question.
- A related questions from above, i understand the folding (RQ4) task for the authors is just a “proof-of-concept” such that the authors adopt downsampling for the openfold training data. However, 1.5% of protein single chains can have large variance for the model to show enough performance bias. Could the authors test the robustness of this downsampling strategy and show error bar on it?
- Question regarding the results incorporating the protein-protein interaction PPI graph saying LC-PLM-G.  In specific, in table 3 of the two tasks from TAPE benchmark, the LC-PLM-G/ESM-2-G shows very similar results compared to their vanilla version (LC-PLM/ESM-2). The performance improvement margin seems to be within the reported std range and I do not see significant improvement of doing this; in table 4/ProteinGym benchmark, for ESM-2-G, the incorporating of PPI graph drastically hurt the performance on top of ESM-2 while the performance gap between LC-PLM-G/LC-PLM is also hard to tell. These results can weaken the “necessity” claim of the main motivation of this paper: modeling based on multiple sequence input (PPI in this case). Could the author further justify the benefit for inference with additional PPI graph?

- Could the authors mentioned some concrete modeling examples with biological significance that we have to turn to Mamba because the attention/Transformer sucks or even fails to handle the problem at all? Can the authors provide further evidence that the extended context window length improves the “related” downstream protein tasks beyond the results presented? At present, it is still unclear to me why should I buy the idea of using S4/mamba instead of attention?

MISC:
- In figure 6, why using “value x 2^1” as the ticks? That looks a bit weird

---

> ### Author Response · Authors · 2024-11-22
> **Response to Reviewer 8QMc [1/n]**
>
> We thank the reviewer for their valuable feedback on our paper. We performed multiple novel ablations and comparisons that we report in the main comment of the rebuttal. We also address the reviewer’s specific comments and concerns below:
>
> >**1.Weak motivation in using SSMs: no latency constraints for pLMs.**
>
> - We want to emphasize and restate our main motivation for using BiMamba-S as our architectural design choice in our work – our LC-PLM focuses on **learning a long-context foundation pLM** that can provide universal amino acid level protein representations for extremely long protein sequences; protein complexes, multimers, and heterodimers with encoded protein interaction context information. SSMs grant the model (1) long-context capability, (2) length extrapolation capability, and (3) high-resolution data modeling capability to realize our goal. In contrast, Transformers have to be retrained with adjusted positional encodings to extrapolate over length [[Liu et al.](https://www.google.com/url?sa=t&source=web&rct=j&opi=89978449&url=https://arxiv.org/abs/2312.17044&ved=2ahUKEwi-iMyxxPCJAxW4kokEHQNYHhcQFnoECCEQAQ&usg=AOvVaw0fldeIoSBJQd5fNgnKjOAN)] or can not perform well on high-resolution data [[Wang et al.](https://arxiv.org/abs/2401.13660), [Schiff et al.](https://arxiv.org/abs/2403.03234)].
>
> - Along with such motivations, we empirically demonstrate that LC-PLM has improved length extrapolation capabilities, favorable scaling laws, and achieved a 7% to 34% improvement on downstream tasks (e.g. protein structure prediction (CASP15-multimers, CASP14, Benchmark2), tasks in TAPE and ProteinGym) compared to ESM-2 [[Lin et al.](https://www.biorxiv.org/content/10.1101/2022.07.20.500902v1.full.pdf)] and CARP [[Yang et al.](https://www.biorxiv.org/content/10.1101/2022.05.19.492714v2.full.pdf)], especially for longer proteins and protein complexes; (2) to encode biological interaction information, we propose a novel second-stage training based on random walks over graphs to extend the long-context capabilities of LC-PLM to leverage the protein interaction graph context; (3) we demonstrate its effectiveness in capturing graph contextual information on remote homology detection (TAPE) in Table 3, protein function prediction (ogbn-proteins) in Table 14 and 15, and PPI link prediction (ogbl-ppa) in Table 16 and 17.
>
> - In addition to our main motivations above, we argue that the inference efficiency is important in computational protein design/drug discovery applications: one usually generate up to $10^6$ protein sequences as candidates [[Adolf-Bryfogle et al.](https://pubmed.ncbi.nlm.nih.gov/29702641/)] and pLMs fine-tuned for scoring those designed protein sequences can be used for scoring, ranking, and filtering the designed protein sequences. The per-step constant time complexity of SSMs could be an advantage in accelerating this phase.
>
> > **2.Weak motivation in long-context modeling of proteins.**
>
>
> - We thank the reviewer for this comment. We clarify the biological motivations and needs for long-context modeling of proteins into three perspectives: (1) functional, (2) structural, and (3) evolutionary. (1) many proteins function as part of multi-protein complexes (e.g. transcription factors) and physically or functionally interact with other proteins and molecules. The interaction information is often captured in protein-protein interaction graphs. Knowing the interacting partners of an individual protein is helpful in predicting the protein’s properties. We demonstrate this on protein function prediction tasks using the ogb-proteins graphs, as shown in Tables 14 and 16. With interaction information, the model shows performance gain. (2) Protein structure depends on global fold, which can involve residues and interactions across long distances and across multiple protein sequences. Modeling multi-protein systems captures distant dependencies critical for stability and function. Folding of multi-meric protein complexes relies on models capable of handling long contexts. We demonstrate this benefit in our LMFold experiments in Table 2. Our model outperforms ESM-2 across all folding benchmarks, especially on CASP15-multimers. (3) Proteins in the same pathway or family exhibit co-evolutionary patterns due to functional interdependencies. In fact, multiple sequence alignment (MSA) of homologous protein sequences is a common approach to increase the contexts for studying individual proteins. As other reviewers noted, ProtMamba [[Sgarbossa et al.](https://www.biorxiv.org/content/10.1101/2024.05.24.595730v1)] is inspired by leveraging MSA as an individual protein’s context.

---

> ### Author Response · Authors · 2024-11-22
> **Response to Reviewer 8QMc [2/n]**
>
> >**3. Compute-efficient Transformers.**
>
> - As we discussed in the above first and second points, our main goal is not to design a compute-efficient sequence modeling architecture. We emphasize other perspectives that SSMs / linear RNNs over Transformers in learning a pLM with better performances. This compute-efficient feature of SSMs is a gifted advantage that we naturally gain from using such a design choice. We also want to note that Flash Attention is a hardware-efficient implementation of Transformers that **cannot reduce the training and inference time complexity of Transformers**. It only provides faster implementation in terms of wall-clock time. Mamba currently also has decent hardware-efficient implementation [[Dao & Gu](https://www.google.com/url?sa=t&source=web&rct=j&opi=89978449&url=https://arxiv.org/abs/2405.21060&ved=2ahUKEwiEuaD_xfCJAxU1EVkFHcdFIUUQFnoECBoQAQ&usg=AOvVaw2pz2eYNg_qM7rYXQYQIIqH),[Megatron](https://github.com/NVIDIA/Megatron-LM/blob/main/pretrain_mamba.py)] such as parallelized associative scan [[Gu & Dao](https://www.google.com/url?sa=t&source=web&rct=j&opi=89978449&url=https://arxiv.org/abs/2312.00752&ved=2ahUKEwjK6p2axvCJAxUQMlkFHYBLAyoQFnoECAwQAQ&usg=AOvVaw3crj6SFh5WpnEaozDiZhbi)] to make it more efficient in terms of wall-clock time. Other efficient Transformers that leverage approximate attention mechanisms (e.g. linear attention) are essentially a reformulation of linear RNNs/SSMs [[Katharopoulos et al.](https://www.google.com/url?sa=t&source=web&rct=j&opi=89978449&url=https://arxiv.org/abs/2006.16236&ved=2ahUKEwiPr4PJxfCJAxVxEVkFHRigOSEQFnoECBUQAQ&usg=AOvVaw2btv9SP2yqT9dOEVjuLQ0G), [Yang et al.](https://www.google.com/url?sa=t&source=web&rct=j&opi=89978449&url=https://openreview.net/pdf%3Fid%3Dia5XvxFUJT&ved=2ahUKEwiPr4PJxfCJAxVxEVkFHRigOSEQFnoECCcQAQ&usg=AOvVaw2HieVMQ_M3Z209OmN0w5o3)].
>
> - We also want to note that there exist many decent linear RNNs/Transformers architectures in the community but our work is not debating on selecting/tailoring a specific architecture. Just like how Transformers have been used in numerous impactful works such as GPT and ESM-2, BiMamba-S also serves as a versatile and effective architectural choice for sequence modeling. The reuse of proven architectures like Transformer or BiMamba does not diminish the novelty of a work. Instead, it allows researchers to focus on solving domain-specific challenges (in our case, exploring the capability of building up a new type of foundation pLM based on BiMamba-S). Our work follows this principle, leveraging this architecture to demonstrate its effectiveness in pushing the boundaries of foundation pLM.
>
> - Although compute-efficient architecture is not our focus, we also want to argue that there is indeed practical demand for compute-efficient pLMs as we discussed above. The inference efficiency of pLMs is important in computational protein design/drug discovery applications: one usually generate up to $10^6$ protein sequences as candidates [[Adolf-Bryfogle et al., 2018](https://pubmed.ncbi.nlm.nih.gov/29702641/)] and pLMs fine-tuned for scoring those designed protein sequences can be used for scoring, ranking, and filtering the designed protein sequences. The per-step constant time complexity of SSMs could be an advantage in accelerating this phase.
>
> >**4. Why not other SOTAs?**
>
> - We want to highlight that our goal is to build up a foundation pLM that can learn meaningful universal amino acid level representations for **pure** protein sequences and generalize across various downstream tasks. As you pointed out, our well-trained pLM **can be used as a drop-in replacement for ESM-2 to accommodate any of these existing methods on their tasks**. In fact, we demonstrated this in practice with the LMFold experiments (a generalization of ESMFold), where a simple folding trunk is trained to predict protein structures based on pLM embeddings. We only show the comparison against existing SOTA open-sourced pLMs trained on protein sequences alone to avoid confounding gains resulting from ad-hoc methods built off pre-trained pLMs. For example, on the ProteinGym leaderboard, SOTA methods like PoET [[Truong Jr et al.](https://arxiv.org/pdf/2306.06156)], TranceptEVE [[Notin et al.](https://openreview.net/forum?id=l7Oo9DcLmR1)] rely on combining family-specific models or alignment-based methods with a foundation pLM; SaProt [[Su et al.](https://www.biorxiv.org/content/10.1101/2023.10.01.560349v5)] is a pretrained pLM with massive protein structure data. We added these SOTA methods to our table but **we want to emphasize again that our work is to develop a foundation pLM (i.e. LC-PLM and LC-PLM-G) that can be utilized as a better pLM backbone in all these works**. And we believe these methods built off our LC-PLM have a high probability of outperforming ESM-2 given all the evidence we provided in our work.

---

> ### Author Response · Authors · 2024-11-22
> **Response to Reviewer 8QMc [3/n]**
>
> >**5. How did we hold out the test 250K? Do we retrain ESM-2 for Figure 4? What is “evaluation loss”?**
>
> - Similar to ESMFold [[Lin et al.](https://www.google.com/url?sa=t&source=web&rct=j&opi=89978449&url=https://www.science.org/doi/10.1126/science.ade2574&ved=2ahUKEwj8yOyRx_CJAxVtD1kFHUdfEgEQFnoECB8QAQ&usg=AOvVaw2uZOq8F6b3Ys4mkbF9t3hQ)], the hold-out 250K sequences are randomly sampled from UniRef90. For the training set, we used `mmseqs` to filter out sequences in the UniRef50 data with >90% sequence identity, such that the remaining sequences in UniRef50 are no more than 90% similar to any of the sequences in the hold-out test set. We provide the details in Appendix B.
> - Yes, we retrained ESM-2 following the [official recipe](https://www.google.com/url?sa=t&source=web&rct=j&opi=89978449&url=https://github.com/facebookresearch/esm&ved=2ahUKEwie9OWcx_CJAxWgF1kFHUKaOAwQFnoECAwQAQ&usg=AOvVaw1qbC62hfVOkizvBbKD47EH) for different sizes using the same train set and test set as we used for LC-PLM.
> - Evaluation loss is the **average cross-entropy across all tokens** (equivalent to [Perplexity (PPL)](https://www.google.com/url?sa=t&source=web&rct=j&opi=89978449&url=https://en.wikipedia.org/wiki/Perplexity&ved=2ahUKEwiVpZ-_x_CJAxWQGFkFHdLyIUYQFnoECBIQAQ&usg=AOvVaw0U47jF3dhfPmb6wd8SKD30)), which is the standard way that people use to evaluate language models . We added this description to our manuscript.
>
> >**6. Align folding trunk in structure prediction.**
>
> In the folding evaluation presented in RQ4, Table2, we are essentially evaluating how well the residue-level embeddings from different pLMs can predict the 3D structures. We perform this evaluation by learning a folding trunk with its architecture adopted from ESMFold for each pLM on paired protein embeddings and ground truth structures, then validate the performance on hold out structure datasets (CASP14, CASP15-multimer, Benchmark2). It is worth noting that the folding chunks for each pLM are trained from scratch rather than transfer learning. This evaluation setting is similar to linear probing, where the parameters in the LMs are frozen and the probing head; in this case, a single folding trunk, is learned by stochastic gradient descent.
>
> >**7. Robustness of downsampling.**
>
> Thanks for this great feedback! We add new results to assess the robustness of downsampling. We perform three 1.5% downsampling using three random seeds and retrain both our LC-PLM and ESM-2. The results are in the table below. We find that the downsampling is very robust to the performance with a small standard deviation that is close to (even smaller than) the standard deviation of using the same train set but training with different random seeds as we reported in our original table.
>
> | Models                  | CASP15-multimers    | CASP14             | Benchmark2         |
> |-------------------------|---------------------|--------------------|--------------------|
> | ESM-2-650M (100B)       | 0.4132 ± 0.0065    | 0.3437 ± 0.0039    | 0.4773 ± 0.0092    |
> | LC-PLM-790M (100B)      | 0.5004 ± 0.0139    | 0.4244 ± 0.0053    | 0.6290 ± 0.0121    |
> | ESM-2-public (1T, for reference only)        | 0.5128 ± 0.0003| 0.4421 ± 0.0023| 0.6844 ± 0.0059|
>
> >**8. Usefulness of graph context.**
>
> Thanks for bringing up this concern! We want to refer to Tables 14, 15, 16, 17 to show that the encoded graph context information can help with protein function prediction and protein interaction prediction. But to make this claim much clearer, we also add more experiments (as shown in the table below) on downstream tasks to verify the effectiveness of the graph contextual training. LC-PLM-G outperforms its vanilla variant on 3/4 TAPE tasks, as shown in the table below. We also want to note that by comparing two LC-PLM-G variants trained on different PPI graphs, the performance also varied a lot, which indicates that the data quality of the PPI graph is also important for the performance boost. We think building up a high-quality PPI graph can be meaningful future work that makes the pretrained pLM better. Regarding the fact that incorporating PPI graphs drastically hurts the performance of ESM-2, this is potentially due to **the poor length extrapolation capability (as shown in Figure 6) and the catastrophic forgetting issue** [[Kenneweg et al.](https://arxiv.org/abs/2404.01317), [Luo et al.](https://arxiv.org/abs/2308.08747)] of Transformers. In Figure 6, we show that if we train ESM-2 on longer sequences, the model will fail to extrapolate on both shorter and longer sequences. Thus, after the second-phase graph context training, ESM-2 forgets the high-quality representations for regular-length protein sequences (shorter than graph-contextualized sequences) learned in the first-phase pretraining and fails to extrapolate on these shorter sequences. It can only provide degenerated representations of them.

---

> ### Author Response · Authors · 2024-11-22
> **Response to Reviewer 8QMc [4/n]**
>
> | Models               | PPI Graph       | Jacobian Contact Map | Remote Homology     | Secondary Structure | Stability       | Fluorescence    |
> |-----------------------|-----------------|-----------------------|---------------------|---------------------|-----------------|-----------------|
> | LC-PLM-790M (100B)   | None            | 47.1                 | 35.14 ± 1.69        | **85.07 ± 0.03**    | 0.794 ± 0.003 | 0.692 ± 0.002   |
> | LC-PLM-G-790M (100B) | ogbn-proteins   | 47.15                | **35.74 ± 0.93**        | 85.02 ± 0.11    | **0.801 ± 0.001** | **0.709 ± 0.033**   |
> | LC-PLM-G-790M (100B) | ogbl-ppa        | **47.23**            | 35.60 ± 1.45        | 85.01 ± 0.03    | **0.801 ± 0.001** | 0.693 ± 0.002   |
>
> >**9. Concrete modeling examples with biological significance that we have to turn to Mamba because the attention/Transformer sucks or even fails to handle the problem at all? Can the authors provide further evidence that the extended context window length improves the “related” downstream protein tasks beyond the results presented?**
>
> - We appreciate this comment and address reviewer’s concern in twofold:
>
> --- (1) LC-PLM has favorable adaptability to long-context tuning: We performed three more downstream protein tasks to evaluate ESM2 and LC-PLM models. In our existing and new experiments (compiled in Table A), after performing the 2nd stage graph training, the performance of ESM2 (Transformers) degrades on many downstream tasks including protein fitness prediction in ProteinGym (Table 4, Spearman’s rho drop from 0.295 to 0.109), Contact map prediction (New table below, precision drops from 44.1 to 26.7), and TAPE stability prediction (New table below, Spearman’s rho drop from 0.763 to 0.750). On the other hand, our LC-PLM models maintain or slightly improve their performances on these tasks after the 2nd stage graph training. These results suggest it is difficult to tune Transformer models to adapt to longer contexts.
>
> --- (2) Mamba-based models enjoy favorable inference efficiency in terms of both time and space complexity. This will satisfy the practical demands for in-silico protein design, where one needs to screen $10^6$ sequences using pLM based methods. The constant time complexity of Mamba/SSMs could be an advantage in accelerating this phase. There is also a GPU memory constraint in performing inference with the Transformer/ESM2 model on long protein sequences users of the ESM2 model have been facing [[issue1](https://github.com/facebookresearch/esm/issues/21), [issue2](https://github.com/facebookresearch/esm/issues/49)].
>
> Table A: Evaluation of pLMs before and after 2nd stage graph context training on downstream tasks. For the Jacobian contact map prediction task, we adopted the methods from [[Zhang et al.](https://www.biorxiv.org/content/10.1101/2024.01.30.577970v1)] to use categorical Jacobian matrices computed from protein language models as the zero-shot prediction for protein contact maps and report the precision@2/L (L is the length of a protein sequence) on the validation set of ProteinNet dataset [[AlQuraish](https://bmcbioinformatics.biomedcentral.com/articles/10.1186/s12859-019-2932-0)]. We report the Spearman’s correlation coefficients for the test sets for the TAPE Stability prediction tasks and ProteinGym DMS substitutions benchmarks.
>
> | Models               | PPI Graph       | Jacobian Contact Map | TAPE Stability     | ProteinGym DMS substitutions |
> |-----------------------|-----------------|-----------------------|--------------------|-----------------------------|
> | ESM-2-650M (100B)    | None            | 44.05                | 0.763 ± 0.008      | 0.295 ± 0.013              |
> | ESM-2-G-650M (100B)  | ogbn-proteins   | 32.35                | 0.750 ± 0.016      | 0.109 ± 0.013              |
> | ESM-2-G-650M (100B)  | ogbl-ppa        | 26.66                | 0.753 ± 0.009      | 0.131 ± 0.014              |
> | LC-PLM-790M (100B)   | None            | 47.1                 | 0.794 ± 0.003      | 0.378 ± 0.008              |
> | LC-PLM-G-790M (100B) | ogbn-proteins   | 47.15                | **0.801 ± 0.001**      | **0.380 ± 0.008**              |
> | LC-PLM-G-790M (100B) | ogbl-ppa        | **47.23**                | **0.801 ± 0.001**      | **0.380 ± 0.008**              |
>
> >**10. Weird y-ticks.**
>
> Thanks for this comment. The $2^1$ is due to the log scale on y-ticks. We fixed it in our manuscript to the normal scale.

---

> ### Author Response · Authors · 2024-11-25
> **Kindly reminder of the end of rebuttal**
>
> Dear Reviewer 8QMc,
>
> As the discussion period is coming to an end tomorrow, we kindly ask you to review our response to your comments and let us know if you have any further queries. Alternatively, if you think we addressed your concerns properly and could raise the rating of the paper, we would be extremely grateful. We eagerly anticipate your response and are committed to addressing any remaining concerns before the discussion period concludes.
>
> Best regards,
>
> Authors

---

> > ### Comment · Reviewer_8QMc · 2024-11-29
> > **Further response by Reviewer 8QMc**
> >
> > Thank you for addressing my prior comments. To wrap up my thoughts and inspire further improvements to the manuscript, I would like to propose the following points mainly concerning me.
> >
> > **1. Problem Tackled**
> >
> > - The manuscript appears to propose a new type of foundational protein language model (pLM), LC-PLM, intended to be parallel to existing models like the ESM series (eg. ESM-2). The proposed pLMs are expected to address biologically relevant problems more effectively than these baselines, especially ESM-2 (650M), as emphasized by the authors.
> > - However, across the tasks and results presented, LC-PLM does not demonstrate parity with or superiority over ESM-2. Foundational problems are ambitious and potentially impactful when tackled effectively, but also difficult. I suggest the authors focus on a specific problem where long-context modeling is essential and where ESM-2 underperforms. Demonstrating such a use case would significantly strengthen the contribution.
> >
> > **2. Motivation**
> >
> > The motivation for integrating Mamba/SSM into a pLM requires clearer articulation. While the authors provided “three points” ((1) functional, (2) structural, and (3) evolutionary) in the rebuttal, I remain unconvinced from both biological side and ML side.
> >
> > - The main selling point of Mamba in text/NLP domains is its long-context capability, which is acknowledged. However, its extension to the protein domain requires stronger and specific justification. PPI graph in my opinion is not a persuasive case.
> > - The transition from an ESM model to LC-PLM feels unmotivated for potential users. While the authors may have confidence in the framework, others still need a clear and compelling reason for this shift. I recommend reorganizing the introductory sections to restate the motivation and explicitly address the benefits of Mamba-based pLMs for proteins.
> >
> > **3. Support for Claims**
> >
> > The experimental results presented fail to convincingly support the proposed architecture as an alternative for the specified tasks. Specific concerns include:
> >
> > - Retraining with limited training tokens and applying graph augmentation, which appear to disproportionately affect ESM performance, making the comparison feel unfair.
> > - The lack of scenarios where LC-PLM demonstrates clear advantages over the transformer-based ESM. Providing rigorous and unbiased experiments would strengthen the manuscript.
> >
> > **4. Significance**
> >
> > Although SotA performance is not a mandatory criterion for acceptance, the current manuscript lacks sufficient evidence of new knowledge or value for the community.
> >
> > - It is acknowledged that Mamba/SSM excels in long-context input, including protein sequences. However, to establish significance, the authors could either propose new learning algorithms to address foundational protein modeling challenges or demonstrate LC-PLM’s advantages in specific yet biologically meaningful scenarios where ESM-2 underperforms.
> > - Claiming LC-PLM superiority over transformer-based models without robust experimental support risks misleading the community and detracts from the potential impact of the work.
> >
> >
> > I maintain my original rating and do not recommend acceptance at this time. The manuscript requires stronger motivation, clearer contributions to the community (and optionally more compelling results) to reach its full potential. That being said, I think LC-PLM can potentially become a good work after proper and careful revision :)

---

> ### Author Response · Authors · 2024-11-29
> **Response to Reviewer 8QMc**
>
> We acknowledge and commend the reviewer's well-structured feedbacks for further improving our manuscript. We commit to incorporate those reasonable and constructive comments in the next version of our manuscript after the review period. While some of the comments are objective and reasonable, we do want to respond to a few points that are unfair and self-contradictory:
>
> > 1. Problem Tackled
> > However, across the tasks and results presented, LC-PLM does not demonstrate parity with or superiority over ESM-2.
>
>
> Throughout the experiments presented in our manuscript, we benchmark LC-PLM against ESM-2 we pretrained with the exact same amount of tokens to demonstrate clear performance advantage of LC-PLM over ESM-2. This comparison is intended to **rigorous** control for the variabilities in the quality and quantity of pretraining dataset, such that the observed performance differences can be **unbiasedly** attributed to architectural advantages. On the other hand, it would be unfair to compare models pretrained with different datasets to draw conclusion about the architectural superiority.
>
> > 2. Motivation
> > PPI graph in my opinion is not a persuasive case.
>
>
> It would be extremely helpful for the reviewer to articulate the reason why PPI graph is not a persuasive case for long-context capability, or perhaps suggest a more persuasive case for proteins' long-context use case/applications. From a systems biology perspective, PPI graphs capture functional contexts for individual proteins to help us understand how proteins function in cells/biological systems.
>
> > 3. Support for Claims
> > Retraining with limited training tokens and applying graph augmentation, which appear to disproportionately affect ESM performance, making the comparison feel unfair.
>
>
> As mentioned in the response to the reviewer's point 1, the purpose of comparing LC-PLM and ESM2 with the same amount of training tokens is to control for the variabilities in pretraining data. The graph augmentation training for both LC-PLM and ESM2 were also performed under the exact same training data and procedure. The observed advantage of LC-PLM over ESM2 in controlled setting helps us tease out the confounding effects from differences in pretraining data. It would be helpful for the reviewer to clarify why such comparison "_feel_" unfair. We also respectively suggest the reviewer to objectively assess scientific works rather than using subjective views.
>
>
> > 4. Significance.
> > However, to establish significance, the authors could either propose new learning algorithms to address foundational protein modeling challenges or demonstrate LC-PLM’s advantages in specific yet biologically meaningful scenarios where ESM-2 underperforms.
>
> In our manuscript, we demonstrate LC-PLM's advantage over ESM-2 on graph augmentation scenario, which is biological meaningful as PPI graphs, which can be generalized to biological knowledge graphs, are valuable tools to study how proteins function in actual biological systems rather than in isolation.

---

### Official Review · Reviewer_sY9s · 2024-11-03

**Soundness:** 2
**Presentation:** 2
**Contribution:** 2
**Rating:** 3
**Confidence:** 2

**Summary:**

The authors present a protein model which is based on the Mamba backbone which runs in bidirectional mode with parameter sharing. The authors compare their method to ESM-2 for the pre-training data (UniRef90) and for downstream structure prediction tasks. For protein protein interaction tasks, the authors present a graph-based version of their model.

**Strengths:**

- The results indicate that the proposed model outperforms the ESM-2 baseline on both pretraining MLM objective and the downstream tasks.
- The authors perform experiments for a wide range of model sizes ranging from 130M parameters to 1400M parameters.

**Weaknesses:**

- **Little novelty**:
  * Protein modeling with Mamba based backbone architecture is done here [1]. Notably, in a sense [1] also allows for bidirectional information sharing by their fill-in-the-middle objective.
  * Bidirectional Mamba blocks with parameter sharing was already done here [2].

- **Lack of comparison**:
  * If the authors thought, their work is substantially different from [1], they have to compare to [1] and discuss benefits/differences.
  * For all experiments, e.g. ProteinGym experiment (Table 4), SOTA methods and their performances should be reported along with ESM-2 and LC-PLM-*.

- Figure 2: Mistake wrt the orientation of the third linear projection block.



[1] ProtMamba: a homology-aware but alignment-free protein state space model

[2] Caduceus: Bi-Directional Equivariant Long-Range DNA Sequence Modeling

**Questions:**

- Wrt the PPI section: It seems a bit counterintuitive to force the graph-structure like information into tokenized text only to be able to naively apply a language model to it. Why shouldn't it be possible to simply train a GNN on the given data?

---

> ### Author Response · Authors · 2024-11-22
> **Response to Reviewer sY9s [1/n]**
>
> We thank the reviewer for their valuable feedback on our paper. We performed multiple novel ablations and comparisons that we report in the main comment of the rebuttal. We also address the reviewer’s specific comments and concerns below:
>
> >**1. Little novelty and lack of comparison.**
>
> Thank you for this feedback. While we appreciate the recognition of our work’s relation to previous studies, we would like to elaborate on what we wanted to show from experimental comparisons. We also would like to clarify several key distinctions:
>
> - We have discussed the key differences between our work and ProtMamba [[Sgarbossa et al.](https://www.biorxiv.org/content/10.1101/2024.05.24.595730v1)] in lines 152-153 and 158-161 and Table 1. To summarize, **ProtMamba trained on concatenated homologous protein sequences with autoregressive causal language modeling and an infilling objective focusing on protein sequence generation**. However, our LC-PLM focuses on learning a foundation-level long context protein language model (pLM) that can provide universal amino acid level protein representations for extremely long protein sequences; protein complexes, multimers, and heterodimers with encoded protein interaction context information. We also note that  **LC-PLM is a much better foundation pLM** than the previous open-sourced SOTA foundation pLMs (i.e. ESM-2 [[Lin et al.](https://www.biorxiv.org/content/10.1101/2022.07.20.500902v1.full.pdf)] and CARP [[Yang et al.](https://www.biorxiv.org/content/10.1101/2022.05.19.492714v2.full.pdf)]). We additionally indicate that, unlike ProtMamba, which uses homologous sequences as context, the proposed LC-PLM-G showed a novel way to encode protein interaction graph information that contextualizes functionally related proteins rather than semantically similar ones (i.e. proteins with similar sequences) .
>
> - We thank the reviewer for suggesting a direct comparison with ProtMamba. We want to highlight again that our work aims to build a better protein foundation model instead of a task-specific protein model. We provide new experimental results comparing LC-PLM to ProtMamba. In these new experiments, we evaluate ProtMamba on the downstream tasks we used in our paper. **The performance of ProtMamba is much lower than LC-PLM (even CARP and ESM-2 as shown in general response) across all tasks**. This suggests that ProtMamba pretrained with concatenated homologous protein sequences potentially leads to degraded representations of individual proteins. After all, ProtMamba is trained to use homologous sequences as context for protein generation tasks (e.g., infilling and fitness prediction), rather than producing useful representations for proteins. Also it is worth noting that **ProtMamba cannot extrapolate to sequence > 2048** since they use positional encodings with fixed length in training. We summarized the results in the tables below. We added these results to our manuscript. Also as a side note, ProtMamba (https://openreview.net/forum?id=BMfHO2lXGe) is a concurrent submission to ICLR 2025 so other submissions can be excused for not comparing against them according to ICLR’s policy (https://iclr.cc/Conferences/2025/FAQ).

---

> ### Author Response · Authors · 2024-11-22
> **Response to Reviewer sY9s [2/n]**
>
> Table A. Protein structure prediction with LMFold. Structure prediction performance (TM score) are reported on different hold out datasets.
> | Models               | CASP15-multimers                               | CASP14                               | Benchmark2                         |
> |-----------------------|-----------------------------------------------|--------------------------------------|------------------------------------|
> | LC-PLM-790M (100B)   | **0.5109 ± 0.0070**                           | **0.4154 ± 0.0080**                 | **0.6290 ± 0.0071**               |
> | ProtMamba-public     | N/A (ProtMamba cannot run on protein sequences with > 2048 length) | 0.3288 ± 0.0091                 | 0.4515 ± 0.0062               |
>
> Table B.  Evaluation on TAPE tasks in supervised fine-tuning setting. We report the top-1 accuracy for the Remote Homology fold-level test set; accuracy for the 3-class secondary structure prediction on the CB513 test set; Spearman’s correlation coefficients for the test sets for the Stability and Fluorescence prediction tasks. For the Jacobian contact map prediction task, we adopted the methods from [[Zhang et al.](https://www.biorxiv.org/content/10.1101/2024.01.30.577970v1)] to use categorical Jacobian matrices computed from protein language models as the zero-shot prediction for protein contact maps and report the precision@2/L (L is the length of a protein sequence) on the validation set of ProteinNet dataset [[AlQuraishi](https://bmcbioinformatics.biomedcentral.com/articles/10.1186/s12859-019-2932-0)].
> | Models               | PPI Graph       | Jacobian Contact Map | Remote Homology     | Secondary Structure | Stability       | Fluorescence    |
> |-----------------------|-----------------|-----------------------|---------------------|---------------------|-----------------|-----------------|
> | LC-PLM-790M (100B)   | None            | 47.1                 | 35.14 ± 1.69        | **85.07 ± 0.03**    | 0.794 ± 0.003 | 0.692 ± 0.002   |
> | LC-PLM-G-790M (100B) | ogbn-proteins   | 47.15                | **35.74 ± 0.93**        | 85.02 ± 0.11    | **0.801 ± 0.001** | **0.709 ± 0.033**   |
> | LC-PLM-G-790M (100B) | ogbl-ppa        | **47.23**            | 35.60 ± 1.45        | 85.01 ± 0.03    | **0.801 ± 0.001** | 0.693 ± 0.002   |
> | ProtMamba-public     | None            | 10.96                | 17.82 ± 1.85        | 68.43 ± 0.06        | 0.726 ± 0.012    | 0.688 ± 0.005   |

---

> ### Author Response · Authors · 2024-11-22
> **Response to Reviewer sY9s [3/n]**
>
> - We agreed that Cadeucus [[Schiff et al.](https://arxiv.org/abs/2403.03234)] used a similar architectural design choice and we did cite them and other papers using BiMamba in lines 219-221. However, we want to note that we used BiMamba-S since it is the most proper way to realize bi-directionality in Mamba. **BiMamba-S can be formulated in a more theoretical way as structured SSMs with quasi-separable matrices [[Hwang et al.](https://arxiv.org/abs/2407.09941)]**. We did not apply quasi-separable mixers in our model since Mamba currently has a much **better software-hardware interface environment and distributed training support** in practical implementations that help us train a large foundation-level model feasibly and efficiently. We added this discussion in our paper as well.
>
> - We also want to note that, just like Transformer, which has been used in numerous impactful works such as GPT and ESM-2, BiMamba-S also serves as a versatile and effective architectural choice for long-context sequence modeling. The reuse of proven architectures like Transformer or BiMamba does not diminish the novelty of a work. Instead, it allows researchers to focus on solving domain-specific challenges (in our case, exploring the capability of building up a new type of foundation pLM based on BiMamba-S). Our work follows this principle, leveraging this architecture to demonstrate its effectiveness in pushing the boundaries of foundation pLM.
>
> - Besides the architectural choice based on Mamba, we have **many other important contributions to this field as summarized in the Introduction**: (1) method to encode biological interaction information into pLM, we propose a novel second-stage training based on random walks over graphs to extend the long-context capabilities of LC-PLM to leverage the PPI graph context; (2) we demonstrate that LC-PLM has improved length extrapolation capabilities, favorable scaling laws, and achieved a 7% to 34% improvement on downstream tasks (e.g. protein structure prediction (CASP15-multimers, CASP14, Benchmark2), tasks in TAPE and ProteinGym) compared to ESM-2, CARP, especially for longer proteins and protein complexes; (3) we demonstrate its effectiveness in capturing graph contextual information on remote homology detection (TAPE) in Table 3, protein function prediction (ogbn-proteins) in Table 14 and 16, and PPI link prediction (ogbl-ppa) in Table 15 and 17.
>
> - We want to highlight that our goal is to build up a foundation pLM that can learn meaningful universal amino acid level representations for **pure** protein sequences and generalize across various downstream tasks. That said, for ProteinGym and etc., it is not fair to compare our method to the other ad-hoc methods built off pre-trained pLMs. For example, on the ProteinGym leaderboard, SOTA methods like PoET [[Truong Jr et al.](https://arxiv.org/pdf/2306.06156)], TranceptEVE [[Notin et al.](https://openreview.net/forum?id=l7Oo9DcLmR1)] rely on combining family-specific models or alignment-based methods with a foundation pLM; SaProt [[Su et al.](https://www.biorxiv.org/content/10.1101/2023.10.01.560349v5)] is a pretrained pLM with massive protein structure data. We added these SOTA methods to our table but **we want to emphasize again that our work is to develop a foundation pLM (i.e. LC-PLM and LC-PLM-G) that can be utilized as a better pLM backbone in all these works**. Such ad-hoc design built off the base pLM to improve on some specific task is beyond the scope of our paper. Our original table **only compares foundation pLMs trained on protein sequences alone**.

---

> ### Author Response · Authors · 2024-11-22
> **Response to Reviewer sY9s [4/n]**
>
> >**2.Why don't we simply train a GNN?**
>
> - We want to note that, by training with graph context, we are not aiming to only perform graph tasks (e.g. node prediction, link prediction, etc.), but also want to show that **a good representation with encoded protein interaction context information helps many other downstream tasks (shown in Tables 3, 4, and 12) like remote homology prediction**. We also trained GNNs on other graph-related tasks like ogbn-proteins and ogbl-ppa and conducted ablation studies with and without our learned protein embeddings, as shown in Tables 16 and 17. We show that with better protein representations and improved graph context-aware representations, the performance can be further boosted.
>
> >**3.Wrong direction in the figure.**
>
> - Thanks for the feedback and pointing this out. We have fixed it in our paper.

---

> ### Author Response · Authors · 2024-11-25
> **Kindly reminder of the end of rebuttal**
>
> Dear Reviewer sY9s,
>
> As the discussion period is coming to an end tomorrow, we kindly ask you to review our response to your comments and let us know if you have any further queries. Alternatively, if you think we addressed your concerns properly and could raise the rating of the paper, we would be extremely grateful. We eagerly anticipate your response and are committed to addressing any remaining concerns before the discussion period concludes.
>
> Best regards,
>
> Authors

---

> ### Comment · Reviewer_sY9s · 2024-11-28
> **Reviewer's Answer to the Authors**
>
> Dear Authors,
>
> Thank you for your detailed answers. In the following you will find some follow-up thoughts:
>
> **Novelty**:
> To improve clarity regarding the fact that BiMamba-S is not a novel architectural component but was introduced previously and applied in this work to another domain, you might consider moving the BiMamba-S section to the Preliminaries section.
>
> **Goal of Learning a Long-Context Foundation pLM**:
> Thank you for clarifying the focus of your work. With this context in mind, I believe the manuscript requires improvements in clarity, motivation, and focus:
>
> a) Clarity and Motivation: You should define what properties should a protein foundation model have and how could you test them. I guess, showing downstream task performances to demonstrate representation learning capabilities makes sense, just the motivation is missing.
>
> b) Universal Representation Claim: The manuscript states that LC-pLM “learns universal AA token-level protein representations” (line 161). However, this claim is vague, and the term “universal” is not well-defined.
>
> c) Long-Context Motivation: I agree with reviewer 8QMc’s initial comment that the necessity for long-context capabilities isn’t well motivated in the manuscript, despite the fact that Figure 6 is relevant and interesting, and implicitly shows the foundation models’ need to be able to adapt to larger sequences.
>
> d) Multi-Modal Foundation Models: The authors state in one of the review answers that they want to focus on pure sequence foundation models. Will trained models be relevant though? Wouldn’t people just use multi-model models like ESM-3? I guess, one cannot expect the proposed method matches ESM-3 performance. However, it would be important to know how significant differences are.
>
> I appreciate the performance gains demonstrated by LC-pLM in the experiments. However, my concerns regarding architectural novelty remain. If the primary focus is to develop “protein foundation models,” the manuscript would benefit from significant restructuring and rewriting to reflect that focus more clearly.
>
> For these reasons, I will stick to my initial rating.

---

> > ### Author Response · Authors · 2024-11-28
> > **Discussion [2/n]**
> >
> > > 4. Weak motivation of long-context modeling.
> >
> > We thank the reviewer for this comment. We clarify the biological motivations and needs for long-context modeling of proteins into three perspectives: (1) functional, (2) structural, and (3) evolutionary. (1) many proteins function as part of multi-protein complexes (e.g. transcription factors) and physically or functionally interact with other proteins and molecules. The interaction information is often captured in protein-protein interaction graphs. Knowing the interacting partners of an individual protein is helpful in predicting the protein’s properties. We demonstrate this on protein function prediction tasks using the ogb-proteins graphs, as shown in Tables 14 and 16. With interaction information, the model shows performance gain. (2) Protein structure depends on global fold, which can involve residues and interactions across long distances and across multiple protein sequences. Modeling multi-protein systems captures distant dependencies critical for stability and function. Folding of multi-meric protein complexes relies on models capable of handling long contexts. We demonstrate this benefit in our LMFold experiments in Table 2. Our model outperforms ESM-2 across all folding benchmarks, especially on CASP15-multimers. (3) Proteins in the same pathway or family exhibit co-evolutionary patterns due to functional interdependencies. In fact, multiple sequence alignment (MSA) of homologous protein sequences is a common approach to increase the contexts for studying individual proteins. As other reviewers noted, ProtMamba [[Sgarbossa et al.](https://www.biorxiv.org/content/10.1101/2024.05.24.595730v1)] is inspired by leveraging MSA as an individual protein’s context.
> >
> > We also want to refer to Tables 14, 15, 16, 17 to show that the encoded graph context information can help with protein function prediction and protein interaction prediction. But to make this claim much clearer, we also add more experiments (as shown in the table below) on downstream tasks to verify the effectiveness of the graph contextual training. LC-PLM-G outperforms its vanilla variant on 3/4 TAPE tasks, as shown in the table below. We also want to note that by comparing two LC-PLM-G variants trained on different PPI graphs, the performance also varied a lot, which indicates that the data quality of the PPI graph is also important for the performance boost. We think building up a high-quality PPI graph can be meaningful future work that makes the pretrained pLM better. Regarding the fact that incorporating PPI graphs drastically hurts the performance of ESM-2, this is potentially due to **the poor length extrapolation capability (as shown in Figure 6) and the catastrophic forgetting issue** [[Kenneweg et al.](https://arxiv.org/abs/2404.01317), [Luo et al.](https://arxiv.org/abs/2308.08747)] of Transformers. In Figure 6, we show that if we train ESM-2 on longer sequences, the model will fail to extrapolate on both shorter and longer sequences. Thus, after the second-phase graph context training, ESM-2 forgets the high-quality representations for regular-length protein sequences (shorter than graph-contextualized sequences) learned in the first-phase pretraining and fails to extrapolate on these shorter sequences. It can only provide degenerated representations of them.

---

> > ### Author Response · Authors · 2024-11-28
> > **Discussion [3/n]**
> >
> > Comparing ESM-2 (Transformer-based) and LC-PLM (Mamba-based), we have more discussion and evidence to show that the long-context capability is essential for many downstream tasks.
> >
> > - LC-PLM has favorable adaptability to long-context tuning: We performed three more downstream protein tasks to evaluate ESM2 and LC-PLM models. In our existing and new experiments (compiled in Table A), after performing the 2nd stage graph training, the performance of ESM2 (Transformers) degrades on many downstream tasks including protein fitness prediction in ProteinGym (Table 4, Spearman’s rho drop from 0.295 to 0.109), Contact map prediction (New table below, precision drops from 44.1 to 26.7), and TAPE stability prediction (New table below, Spearman’s rho drop from 0.763 to 0.750). On the other hand, our LC-PLM models maintain or slightly improve their performances on these tasks after the 2nd stage graph training. These results suggest it is difficult to tune Transformer models to adapt to longer contexts.
> >
> > - Mamba-based models enjoy favorable inference efficiency in terms of both time and space complexity. This will satisfy the practical demands for in-silico protein design, where one needs to screen $10^6$ sequences using pLM based methods. The constant time complexity of Mamba/SSMs could be an advantage in accelerating this phase. There is also a GPU memory constraint in performing inference with the Transformer/ESM2 model on long protein sequences users of the ESM2 model have been facing [[issue1](https://github.com/facebookresearch/esm/issues/21), [issue2](https://github.com/facebookresearch/esm/issues/49)].
> >
> > Table A: Evaluation of pLMs before and after 2nd stage graph context training on downstream tasks. For the Jacobian contact map prediction task, we adopted the methods from [[Zhang et al.](https://www.biorxiv.org/content/10.1101/2024.01.30.577970v1)] to use categorical Jacobian matrices computed from protein language models as the zero-shot prediction for protein contact maps and report the precision@2/L (L is the length of a protein sequence) on the validation set of ProteinNet dataset [[AlQuraish](https://bmcbioinformatics.biomedcentral.com/articles/10.1186/s12859-019-2932-0)]. We report the Spearman’s correlation coefficients for the test sets for the TAPE Stability prediction tasks and ProteinGym DMS substitutions benchmarks.
> >
> > | Models               | PPI Graph       | Jacobian Contact Map | TAPE Stability     | ProteinGym DMS substitutions |
> > |-----------------------|-----------------|-----------------------|--------------------|-----------------------------|
> > | ESM-2-650M (100B)    | None            | 44.05                | 0.763 ± 0.008      | 0.295 ± 0.013              |
> > | ESM-2-G-650M (100B)  | ogbn-proteins   | 32.35                | 0.750 ± 0.016      | 0.109 ± 0.013              |
> > | ESM-2-G-650M (100B)  | ogbl-ppa        | 26.66                | 0.753 ± 0.009      | 0.131 ± 0.014              |
> > | LC-PLM-790M (100B)   | None            | 47.1                 | 0.794 ± 0.003      | 0.378 ± 0.008              |
> > | LC-PLM-G-790M (100B) | ogbn-proteins   | 47.15                | **0.801 ± 0.001**      | **0.380 ± 0.008**              |
> > | LC-PLM-G-790M (100B) | ogbl-ppa        | **47.23**                | **0.801 ± 0.001**      | **0.380 ± 0.008**              |

---

> ### Author Response · Authors · 2024-11-28
> **Discussion [1/n]**
>
> Thank you for your feedback. Here we provide further discussion.
>
> > 1. No novelty of BiMamba-S; Moving it to Prelim.
>
> We respectfully disagreed.
>
> (1) Note that, as we mentioned in our rebuttal, Cadeucus has similar architectural design choice that is **not exactly same**. Here are several key differences:
> - "shared layers" is not qual to "tied weights". (a) "Shared layers" are more efficient since they reuse the same layer object, whereas weight tying tracks separate layers with identical weights, introducing minor overhead. (b) "Shared layers" simplify optimization as they operate on a single computational graph, while weight tying requires additional constraints to enforce equality during training.
> - We have another design choice "untied input/output embedding layers" and we carefully study it. This can help alleviate collapsed embedding space and improve the uniformity of learned embeddings, which make the model more expressive and avoid *anisotropic* issue.
> - We perform the reverse operation after a normalization layer and we provide a residual connection at the end of the block to help gradient flow.
>
> (2) Also note that Caduceus is not the first one proposing to use bidirectional Mamba (BiMamba). There has been a lot of works built off BiMamba, just as we discussed in our manuscript -- "... time-series forecasting (Liang et al., 2024), audio representation learning (Erol et al., 2024), visual representation learning (Zhu et al., 2024), DNA modeling (Schiff et al., 2024), and graph learning (Behrouz & Hashemi, 2024)." These works came out on various dates but all discussed their BiMamba architecture designs in their model/methodology. We think it's worth to also discuss our own architectural designs in the method part to indicate the similar insights and key differences in a totally different field.
>
> > 2. Clarity and motivation.
>
> Note that we're not proposing a *protein foundation model*; instead, we're introducing another **foundation-level protein language model** (pLM). We strongly recommend the reviewer to read ESM-2 (Lin et al.) to get the motivation of having a foundation-level pLM if they're not familiar with this field and think the motivation is missing.
>
> Also, the community hasn't had a clear definition of "foundation model" and there has been a long debate on this. Thus, it's also not easy for us to define its necessary properties and claim we achieved "foundation" in our work. Given these, we choose to avoid this term but focus on demonstrating we have a better pLM compared to existing works. As we introduced in the manuscript, we think we achieved this by the following evidence:
> - *We demonstrate that LC-PLM has improved length extrapolation capabilities, favorable scaling laws, and achieved a 7% to 34% improvement on downstream tasks (e.g. protein structure prediction (CASP15-multimers, CASP14, Benchmark2), tasks in TAPE and ProteinGym) compared to ESM-2, especially for longer proteins and protein complexes.*
> - *To encode biological interaction information, we propose a novel second-stage training based on random walks over graphs to extend the long-context capabilities of LC-PLM to leverage the PPI graph context. We demonstrate its effectiveness in capturing graph-contextual information on remote homology detection (TAPE), protein function prediction (ogbn-proteins), and PPI link prediction (ogbl-ppa).*
>
> > 3. "Universal Representation" is not well-defined.
>
> We already addressed this concern in our reply to Review jUT5. For your convenience, we restate here again:
>
> We use the term “universal” since (1) this term has been widely-used in protein representation learning literature [[Alley et al.](https://pubmed.ncbi.nlm.nih.gov/31636460/), [Detlefsen et al.](https://www.nature.com/articles/s41467-022-29443-w)], where researchers define pLM is *“[learning universal, cross-family representations of protein space](https://www.nature.com/articles/s41467-022-29443-w)”*; and (2) we pretrain our models on the **Universal** Protein Reference Clusters (UniRef) dataset which contains universal protein sequence resources. Given such learned high-quality protein embeddings, we can achieve decent performance across a variety of downstream tasks, which demonstrates the universality of such protein representations.

---

> ### Author Response · Authors · 2024-11-28
> **Discussion [4/n]**
>
> > 5. Why don't people just use multi-modal models like ESM-3? Why do we still work on pure pLM?
>
> We elaborate the reasons as follow:
> - We kindly note that **ESM-3 is also a pLM** but with training data tokenized from multiple sources including protein sequences, structures, and functions. So if we can build up a better pLM, we can also make a better ("multi-modal") version of LC-PLM in the future with the same training data, which can outperform ESM-3.
> - Also, note that ESM-3 still suffers the same intrinsic issue that it has weak long-context capability of modeling protein complexes, heterodimers, interaction contexts, and etc. Therefore, if our proposed LC-PLM can alleviate this weakness, we will have a better model to adapt to these mentioned scenarios/tasks.
> - Developing a better base model is orthogonal to training a multi-modal model. For example, the researchers are still putting efforts on developing new language models that have other features like highly compute-efficient, capture long-context, etc. Why don't they just use GPT-4o and stop researching on new LMs? Our work follows the same motivation and insights.
>
> Lastly, we kindly ask the reviewer to **carefully read the related literature in this domain and look into our response and manuscript** before making arguments. Thank you.

---

### Official Review · Reviewer_Z2Tu · 2024-11-04

**Soundness:** 4
**Presentation:** 4
**Contribution:** 3
**Rating:** 8
**Confidence:** 3

**Summary:**

The authors suggest protein language models, denoted as LC-PLM and LC-PLM-G, respectively.
LC-PLM is based on a Mamba-based architecture, which they call BiMamba-S.
Two main ideas of BiMamba-S are bidirectionality and shared projection layers for forward and flipped inputs.
Sharing of layers allows deeper layers, since the number of parameters is reduced.
The authors also suggest an extension for knowledge graphs, which leads to LC-PLM-G.
Graphs are represented as random walks between nodes, where nodes are protein sequences themselves and edges are indicated by an EDGE token.
They also use negative random walk samples, where there is no edge between nodes and mark it with a special NO_EDGE token.
The authors apply their methods/models to several downstream task and find superiority of their method over trained versions of ESM2.

**Strengths:**

- relevant topic
- good empirical results
- innovative idea how to integrate knowledge graphs into the models
- well-written paper

**Weaknesses:**

- It would be interesting to see the performance of ProtMamba to be included in comparisons, where it makes sense.
- It would be interesting to additionally see the performance of Contact Prediction, Fluorescence and Stability on the TAPE tasks.
- Criterions for Table 1 should be better specified (e.g., when is a method considered to be universal?)
- line 101: here also "Hochreiter, S. (1991). Untersuchungen zu dynamischen neuronalen Netzen" should be cited.

**Questions:**

- Which context size do you use for ESM-2? Could it have been increased for the experiments you carried out (i.e., did ESM2 and LC-PLM use approximately the same memory?)?
- What hyperparameters did you use for ESM-2? How was hyperparameter selection done for ESM-2 and LC-PLM(-G)?
- RQ3:\
In contrast to some other experiments UniRef50 is used for training. Evaluation is at UniRef90. Why is this the case?
- line 462-464: "As shown in Figure 7, the embeddings from LC-PLM-G captures the graph topology much better than LC-PLM, which aligns with the community detection results."\
What is the criterion to see that LC-PLM-G captures the graph topology much better?
- line 439-440: "This also suggests that, even for average-length protein sequences, long-range dependencies would be useful information and an important feature for protein structure prediction."\
What is the exact indication to come to this conclusion?
- line 533: "LC-PLM achieved 7% to 34% better performance on various downstream tasks." Which task do you exactly mean with 7% and which with 34%?

---

> ### Author Response · Authors · 2024-11-22
> **Response to Reviewer Z2Tu [1/n]**
>
> We sincerely thank the reviewer for the favorable rating and constructive suggestions! To address the weakness and questions:
>
> >**1. Performance comparison with ProtMamba.**
>
> We evaluated the performance of ProtMamba model against ours across seven tasks to find ProtMamba significantly underperforms our models in all 5 tasks we evaluated (structure prediction, Table 2; Contact map, Remote Homology, Secondary Structure, Stability, Table 3;). Our results demonstrate ProtMamba is not good at producing good representations for protein sequences. Its training regime favors protein generation and contextual fitness prediction, which may degrade the representation quality of individual protein sequences in the meantime.
>
> Table A. Protein structure prediction with LMFold. Structure prediction performance (TM score) are reported on different hold out datasets.
> | Models               | CASP15-multimers                               | CASP14                               | Benchmark2                         |
> |-----------------------|-----------------------------------------------|--------------------------------------|------------------------------------|
> | LC-PLM-790M (100B)   | **0.5109 ± 0.0070**                           | **0.4154 ± 0.0080**                 | **0.6290 ± 0.0071**               |
> | ProtMamba-public     | N/A (ProtMamba cannot run on protein sequences with > 2048 length) | 0.3288 ± 0.0091                 | 0.4515 ± 0.0062               |
>
> Table B.  Evaluation on TAPE tasks in supervised fine-tuning setting. We report the top-1 accuracy for the Remote Homology fold-level test set; accuracy for the 3-class secondary structure prediction on the CB513 test set; Spearman’s correlation coefficients for the test sets for the Stability and Fluorescence prediction tasks. For the Jacobian contact map prediction task, we adopted the methods from [[Zhang et al.](https://www.biorxiv.org/content/10.1101/2024.01.30.577970v1)] to use categorical Jacobian matrices computed from protein language models as the zero-shot prediction for protein contact maps and report the precision@2/L (L is the length of a protein sequence) on the validation set of ProteinNet dataset [[AlQuraishi](https://bmcbioinformatics.biomedcentral.com/articles/10.1186/s12859-019-2932-0)].
> | Models               | PPI Graph       | Jacobian Contact Map | Remote Homology     | Secondary Structure | Stability       | Fluorescence    |
> |-----------------------|-----------------|-----------------------|---------------------|---------------------|-----------------|-----------------|
> | LC-PLM-790M (100B)   | None            | 47.1                 | 35.14 ± 1.69        | **85.07 ± 0.03**    | 0.794 ± 0.003 | 0.692 ± 0.002   |
> | LC-PLM-G-790M (100B) | ogbn-proteins   | 47.15                | **35.74 ± 0.93**        | 85.02 ± 0.11    | **0.801 ± 0.001** | **0.709 ± 0.033**   |
> | LC-PLM-G-790M (100B) | ogbl-ppa        | **47.23**            | 35.60 ± 1.45        | 85.01 ± 0.03    | **0.801 ± 0.001** | 0.693 ± 0.002   |
> | ProtMamba-public     | None            | 10.96                | 17.82 ± 1.85        | 68.43 ± 0.06        | 0.726 ± 0.012    | 0.688 ± 0.005   |

---

> ### Author Response · Authors · 2024-11-22
> **Response to Reviewer Z2Tu [2/n]**
>
> >**2. Evaluation of pLMs on the other 3 TAPE tasks: Contact Prediction, Fluorescence and Stability.**
>
> We appreciate this suggestion and performed evaluations for all the models considered in this manuscript (LC-PLM, LC-PLM-G, ESM-2, ESM-2-G, CARP, ProtMamba). The results are shown in Table A and added to the revised version of our paper. It is worth noting that we adopt a different setting for the Contact Map prediction task for a fair comparison of attention-free models including CARP, LC-PLM, and ProtMamba. Instead of using the attention maps from Transformer-based pLMs to predict the contact maps [[Rao et al. 2020](https://www.biorxiv.org/content/10.1101/2020.12.15.422761v1)], we followed [[Zhang et al. (2024)](https://www.biorxiv.org/content/10.1101/2024.01.30.577970v1)] to use categorical Jacobian matrices computed from protein LMs as the zero-shot prediction for protein contact maps and report the precision@2/L (L is the length of a protein sequence) on the validation set of ProteinNet dataset [[AlQuraishi 2019](https://bmcbioinformatics.biomedcentral.com/articles/10.1186/s12859-019-2932-0)]. In Contact map and Stability prediction tasks, we noted a similar trend where LC-PLM outperforms ESM-2 with the same number of pretraining tokens, and that 2nd stage graph pretraining slight improve LC-PLM while hurting the performance of ESM-2, highlighting the difficulty of adopting ESM-2 to handle longer contexts. For Fluorescence tasks, we noted all models saturated at Spearman correlation coefficient around 0.69. This observation was also made by others [[Rao et al. 2019](https://www.biorxiv.org/content/10.1101/676825v1); [McDermott et al. 2023](https://www.nature.com/articles/s42256-023-00647-z); [Schmirler et al. 2024](https://www.nature.com/articles/s41467-024-51844-2)].
>
> | Models                  | PPI Graph        | Jacobian Contact Map | Stability          | Fluorescence       |
> |-------------------------|------------------|-----------------------|--------------------|--------------------|
> | ESM-2-650M (100B)       | None             | 44.05                | 0.763 ± 0.008      | 0.695 ± 0.002      |
> | ESM-2-G-650M (100B)     | ogbn-proteins    | 32.35                | 0.750 ± 0.016      | 0.694 ± 0.002      |
> | ESM-2-G-650M (100B)     | ogbl-ppa         | 26.66                | 0.753 ± 0.009      | 0.693 ± 0.001      |
> | LC-PLM-790M (100B)      | None             | 47.10                | 0.794 ± 0.003      | 0.692 ± 0.002      |
> | LC-PLM-G-790M (100B)    | ogbn-proteins    | 47.15                | **0.801 ± 0.001**      | **0.709 ± 0.003**      |
> | LC-PLM-G-790M (100B)    | ogbl-ppa         | **47.23**            | **0.801 ± 0.001**      | 0.693 ± 0.002      |
> | ProtMamba              | None             | 10.96                | 0.726 ± 0.012      | 0.688 ± 0.005      |
> | CARP-640M-public       | None             | 25.83                | 0.720 ± 0.010      | 0.680 ± 0.002      |
> | ESM-2-650M-public      | None             | 66.85            | 0.804 ± 0.006  | 0.688 ± 0.001      |

---

> ### Author Response · Authors · 2024-11-22
> **Response to Reviewer Z2Tu [3/n]**
>
> >**3. Criterions for Table 1 should be better specified (e.g., when is a method considered to be universal?)**
>
> We define universality as learning representations are learned to reflect general properties of all proteins, rather than a specific protein family or properties such as post-translational modifications. pLMs satisfying universality can be used as a foundation model to develop specialized methods such as PTM prediction and structure prediction. We added these clarifications to our manuscript.
>
> We use the term “universal” since (1) this term has been widely-used in protein representation learning literature [[Alley et al.](https://pubmed.ncbi.nlm.nih.gov/31636460/), [Detlefsen et al.](https://www.nature.com/articles/s41467-022-29443-w)], where researchers define pLM is *“[learning universal, cross-family representations of protein space](https://www.nature.com/articles/s41467-022-29443-w)”*; and (2) we pretrain our models on the **Universal** Protein Reference Clusters (UniRef) dataset which contains universal protein sequence resources. Given such learned high-quality protein embeddings, we can achieve decent performance across a variety of downstream tasks, which demonstrates the universality of such protein representations.
>
> >**4. line 101: here also "Hochreiter, S. (1991). Untersuchungen zu dynamischen neuronalen Netzen" should be cited.**
>
> Thanks for pointing this out, we added this citation to this paper for their contribution to RNNs.
>
> >**5. Which context size do you use for ESM-2? Could it have been increased for the experiments you carried out (i.e., did ESM2 and LC-PLM use approximately the same memory?)?**
>
> We use 1024 as the context size for the ESM-2 models we trained, which is the same context size used in the public ESM2 models. The sizes remain constant as the sizes of the positional embedding matrix can’t be changed.
>
> >**6. What hyperparameters did you use for ESM-2? How was hyperparameter selection done for ESM-2 and LC-PLM(-G)?**
>
> We adopt the hyperparameter from the official ESM-2 paper [Lin et al. 2023] with some modifications: global batch size=0.5M tokens; peak learning rate = 2e-4, 2000 steps learning rate warm-up, followed by cosine decay schedule; weight decay=0.01; Adam beta1=0.9, beta2=0.98, eps=1e-8; We added those details in our Appendix E.
>
> >**7. RQ3: In contrast to some other experiments UniRef50 is used for training. Evaluation is at UniRef90. Why is this the case?**
>
> Thanks for this question! We and others [[Rives et al. 2019](https://www.google.com/url?sa=t&source=web&rct=j&opi=89978449&url=https://www.pnas.org/doi/10.1073/pnas.2016239118&ved=2ahUKEwiX1Mnu1PCJAxVMEGIAHR7mGQYQFnoECA4QAQ&usg=AOvVaw0GxY6R8v3ee7mp_QSywY2N)] found pretraining with UniRef50 leads to lower perplexity with the same training budget due to the lower-level of similar sequences in UniRef50. However, UniRef90 datasets have sizable proteins across all length bins, which is suitable for the length extrapolation evaluation experiments.
>
> >**8. line 462-464: "As shown in Figure 7, the embeddings from LC-PLM-G captures the graph topology much better than LC-PLM, which aligns with the community detection results. What is the criterion to see that LC-PLM-G captures the graph topology much better?**
>
> We elaborate more on the criterion here. We first run community detection on the graph and label the nodes using its corresponding community labels. Since community detection helps us find local clusters in the graph, this labeling method provides enough information of how the topological structure presented in the graph. Thus, we do t-SNE dimensionality reduction on the learned protein embeddings and visualize the 2-D embeddings in the plot. If closer 2-D embeddings reveal the similar community labels, then our proposed graph context training can be demonstrated as capturing graph relational information well.
>
> >**9. line 439-440: "This also suggests that, even for average-length protein sequences, long-range dependencies would be useful information and an important feature for protein structure prediction. What is the exact indication to come to this conclusion?"**
>
> This conclusion was made based on the performance improvement of LC-PLM on the CASP14 dataset (Table 2). CASP14 contains 37 mostly average-length protein sequences (mean=318.6; median=197).
>
> >**10. line 533: "LC-PLM achieved 7% to 34% better performance on various downstream tasks." Which task do you exactly mean with 7% and which with 34%?**
>
> We found LC-PLM achieved 7% and 34% improvement over ESM-2 on TAPE Secondary Structure (85.07% vs 79.85% in accuracy), and Remote Homology (35.14% vs 26.57% in top-1 accuracy), respectively.
>
> Lastly, we would like to sincerely ask you whether you are willing to increase your confidence score if you think we addressed your comments properly. Thank you so much for your support again!

---

> ### Author Response · Authors · 2024-11-25
> **Kindly reminder of the end of rebuttal**
>
> Dear Reviewer Z2Tu,
>
> As the discussion period is coming to an end tomorrow, we kindly ask you to review our response to your comments and let us know if you have any further queries. Alternatively, if you think we addressed your concerns properly and could raise the rating of the paper, we would be extremely grateful. We eagerly anticipate your response and are committed to addressing any remaining concerns before the discussion period concludes.
>
> Best regards,
>
> Authors

---

> > ### Comment · Reviewer_Z2Tu · 2024-11-29
> >
> > First remarks:
> > - I have to say, that I did NOT read the paper again and also did not carefully check, whether all the points the authors promised are now included.
> > - I have roughly looked over the other reviews and the responses, but not in full detail.
> >
> > with respect to point 3)
> > I am not sure, whether I could find the clarification in the uploaded PDF as they authors did not seem to mark their changes with another color.
> > Especially (with respect to point 3), I did not only mean, that the criterion "universality" needs more accurate definition, but also all the other criterions (Fine granularity, Handleability, Performance, Graph context, Large-scale model). What is the **exact** criterion for a check mark and what is the **exact** criterion that there is no check mark?
> > I think this needs to be clarified, as otherwise it might be unfair to other methods. Please describe as precisely as possible.
> >
> > I will keep my score, as I think it is sufficiently high. I however think the area chair should possibly form their own opinion by having a look through the paper and the discussions here and decide whether other reviewers might have given a too low score.

---

> ### Author Response · Authors · 2024-11-29
> **Response to Reviewer Z2Tu**
>
> We sincerely appreciate your advocacy on our behalf. Your support has been invaluable and deeply meaningful to us.
>
> In our revised manuscript, we have incorporated the points we committed to addressing, making updates in both the main text and the appendix. Additionally, we have highlighted the new additions and adjustments, including the definitions and rationales of specific criterions used in Table 1 (see Appendix A from [here](https://drive.google.com/file/d/1LT_6biW4N2I2vbn6LO8keA1wQWHzLqXe/view?usp=sharing) given currently there is a technical issue on OpenReview for updating the PDF file). Please let us know if there is anything that you feel requires further clarification or adjustment.
>
> Once again, we thank you for your insightful feedback. We kindly ask if you could consider increasing your confidence if you feel we have adequately addressed and clarified the concerns.

---

> > ### Comment · Reviewer_Z2Tu · 2024-12-02
> >
> > I consider confidence as a pure meta-value, how much trust there is in my own review and I don't want to overstate my understanding and knowledge.

---

> > > ### Author Response · Authors · 2024-12-02
> > > **Response to Reviewer Z2Tu**
> > >
> > > Thank you for clarifying your perspective on confidence as a meta-value. That makes total sense to us.
> > >
> > > Again, we greatly appreciate the time and effort you’ve invested in improving our work and your support of our submission. If any remaining aspects of the paper could benefit from further clarification, we would be happy to address them.

---

### Author Response · Authors · 2024-11-22
**General response to all reviewers [1/2]**

We sincerely thank all the reviewers for their constructive feedback. Here, we report a brief summary of common questions across multiple reviewers and results from our new experiments to address them.

> **1. Motivation of long-context pLMs**

The biological motivations for long-context modeling of proteins can be summarized to three aspects:

- Functional: many proteins function as part of multi-protein complexes (e.g. transcription factors) and physically or functionally interact with other proteins and molecules. The interaction information is often captured in protein-protein interaction graphs. Knowing the interacting partners of an individual protein is helpful in predicting the protein’s properties.
- Structural: protein structure depends on global fold, which can involve residues and interactions across long distances and across multiple protein sequences. Modeling multi-protein systems captures distant dependencies critical for stability and function. Folding of multi-meric protein complexes relies on models capable of handling long contexts.
- Evolutionary: Proteins in the same pathway or family exhibit co-evolutionary patterns. Multiple sequence alignment (MSA) of homologous protein sequences is a common approach to increase the contexts for studying individual proteins. As other reviewers noted, ProtMamba [[Sgarbossa et al.](https://www.biorxiv.org/content/10.1101/2024.05.24.595730v1)] is inspired by leveraging MSA as an individual protein’s context.

> **2. Comparison with ProtMamba**

We performed additional evaluation experiments to comprehensively compare the performances of our models with ProtMamba. The new results are summarized in Table A.

Table A: Evaluation of pLMs on downstream tasks. We report the top-1 accuracy for the Remote Homology fold-level test set; accuracy for the 3-class secondary structure prediction on the CB513 test set; Spearman’s correlation coefficients for the test sets for the Stability and Fluorescence prediction tasks. For the Jacobian contact map prediction task, we adopted the methods from [Zhang et al. (2024)](https://www.biorxiv.org/content/10.1101/2024.01.30.577970v1) to use categorical Jacobian matrices computed from protein language models as the zero-shot prediction for protein contact maps and report the precision@2/L (L is the length of a protein sequence) on the validation set of ProteinNet dataset [[AlQuraishi 2019](https://bmcbioinformatics.biomedcentral.com/articles/10.1186/s12859-019-2932-0)]. We report the TM scores for the LMFold structure prediction task.

| Tasks/Models                | ESM-2-650M (100B)   | LC-PLM-790M (100B)    | ProtMamba              |
|-----------------------------|---------------------|-----------------------|------------------------|
| **TAPE**                   |                     |                       |                        |
| Jacobian Contact Map        | 44.05              | **47.1**              | 10.96                 |
| Remote Homology             | 26.57 ± 0.49       | **35.14 ± 1.69**      | 17.82 ± 1.85          |
| Secondary Structure         | 79.86 ± 0.09       | **85.07 ± 0.03**      | 68.43 ± 0.06          |
| Stability                   | 0.763 ± 0.008      | **0.794 ± 0.003**     | 0.726 ± 0.012         |
| Fluorescence                | **0.695 ± 0.002**      | 0.692 ± 0.002     | 0.688 ± 0.005         |
| **LMFold structure prediction** |                 |                       |                        |
| CASP15-multimers            | 0.4228 ± 0.0065    | **0.5109 ± 0.0070**   | N/A *                 |
| CASP14                      | 0.3531 ± 0.0076    | **0.4154 ± 0.0080**   | 0.3288 ± 0.0091       |
| Benchmark2                  | 0.4859 ± 0.0119    | **0.6290 ± 0.0071**   | 0.4515 ± 0.0062       |


We found that our LC-PLM outperforms ProtMamba across all but one task by large margins. These results suggest that ProtMamba pretrained with concatenated homologous protein sequences potentially leads to degraded representations of individual protein sequences. Also it is worth noting that **ProtMamba cannot extrapolate to sequence > 2048 in CASP15-multimer benchmark** since they use positional encodings with fixed length in training. We added these results to our manuscript.

---

> ### Author Response · Authors · 2024-11-22
> **General response to all reviewers [2/2]**
>
> > **3. Distinctions with between LC-PLM and ProtMamba**
>
> There are two key distinctions between our method and ProtMamba:
> - Definition of long contexts for proteins: ProtMamba exclusively use evolutionary contexts of individual proteins in the forms of flattened MSAs, whereas LC-PLM-G primarily use functional and structural contexts of proteins stored in PPI graphs. In fact, LC-PLM-G’s graph contextual method is more generalizable: one can feed the evolutionary contexts in the form of sequence graphs, an alternative representation of MSA [[Benedict et al. 2014](https://www.ncbi.nlm.nih.gov/pmc/articles/PMC4082375)], to train or inference with our LC-PLM-G model. We added these discussions to our manuscript and consider these new directions as future works.
> - Foundation model vs specialized model: our goal is to build a foundation pLM that can learn meaningful representations for protein sequences and generalize across various downstream tasks, including prediction of protein structures, functions, fitness, and interactions. On the other hand, ProtMamba is trained to use homologous sequences as context for protein generation tasks (e.g. infilling and fitness prediction), rather than producing useful representations for protein sequences.
>
> > **4. Benefit of SSM/Mamba-based architecture**
>
> - SSM/Mamba-based models can adapt to long-context training better than Transformer-based architectures
>
> We demonstrate our LC-PLM has favorable adaptability to long-context tuning: We performed three more downstream protein tasks to evaluate ESM2 and LC-PLM models. In our existing and new experiments (compiled in Table B), after performing the 2nd stage graph training, the performance of ESM2 (Transformers) degrades on many downstream tasks including protein fitness prediction in ProteinGym, Contact map prediction, and TAPE stability prediction. On the other hand, our LC-PLM models maintain or slightly improve their performances on these tasks after the 2nd stage graph training. These results suggest it is difficult to tune Transformer models to adapt to longer contexts.
>
> Table B: Evaluation of pLMs before and after 2nd stage graph context training on downstream tasks.
>
> | Models                  | PPI Graph        | Jacobian Contact Map | Stability          | Fluorescence       |
> |-------------------------|------------------|-----------------------|--------------------|--------------------|
> | ESM-2-650M (100B)       | None             | 44.05                | 0.763 ± 0.008      | 0.695 ± 0.002      |
> | ESM-2-G-650M (100B)     | ogbn-proteins    | 32.35                | 0.750 ± 0.016      | 0.694 ± 0.002      |
> | ESM-2-G-650M (100B)     | ogbl-ppa         | 26.66                | 0.753 ± 0.009      | 0.693 ± 0.001      |
> | LC-PLM-790M (100B)      | None             | 47.10                | 0.794 ± 0.003      | 0.692 ± 0.002      |
> | LC-PLM-G-790M (100B)    | ogbn-proteins    | 47.15                | **0.801 ± 0.001**      | **0.709 ± 0.003**      |
> | LC-PLM-G-790M (100B)    | ogbl-ppa         | **47.23**            | **0.801 ± 0.001**      | 0.693 ± 0.002      |
>
> - Practical applications that would benefit from efficient pLMs
>
> Mamba-based models enjoy favorable inference efficiency in terms of both time and space complexity [[Gu & Dao, 2024](https://arxiv.org/abs/2312.00752)]. The inference efficiency is important in computational protein design/drug discovery applications: one usually generate up to 10^6 protein sequences as candidates [[Adolf-Bryfogle et al., 2018](https://pubmed.ncbi.nlm.nih.gov/29702641/)] and pLMs fine-tuned for scoring those designed protein sequences can be used for scoring, ranking, and filtering the designed protein sequences. The per-step constant time complexity of SSMs could be an advantage in accelerating this phase.

---

### Meta-Review · Area_Chair_tRTn · 2024-12-22

**Metareview:**

The reviewers raised major concerns on this paper and most of the issues are not resolved during discussions.

**Additional Comments On Reviewer Discussion:**

There have been extensive discussions, but most of the reviewers are not convinced.

---

### Decision · Program_Chairs · 2025-01-22

Reject